# Intermediate filaments associate with aggresome-like structures in proteostressed *C. elegans* neurons and influence large vesicle extrusions as exophers

Meghan Lee Arnold [1], Jason Cooper[1], Rebecca Androwski [1], Sohil Ardeshna[1], Ilija Melentijevic[1], Joelle Smart[1], Ryan J. Guasp [1], Ken C. Q. Nguyen[2], Ge Bai[1], David H. Hall[2], Barth D. Grant [1]✉ & Monica Driscoll[1]✉

Toxic protein aggregates can spread among neurons to promote human neurodegenerative disease pathology. We found that in *C. elegans* touch neurons intermediate filament proteins IFD-1 and IFD-2 associate with aggresome-like organelles and are required cell-autonomously for efficient production of neuronal exophers, giant vesicles that can carry aggregates away from the neuron of origin. The *C. elegans* aggresome-like organelles we identified are juxtanuclear, HttPolyQ aggregate-enriched, and dependent upon orthologs of mammalian aggresome adaptor proteins, dynein motors, and microtubule integrity for localized aggregate collection. These key hallmarks indicate that conserved mechanisms drive aggresome formation. Furthermore, we found that human neurofilament light chain (NFL) can substitute for *C. elegans* IFD-2 in promoting exopher extrusion. Taken together, our results suggest a conserved influence of intermediate filament association with aggresomes and neuronal extrusions that eject potentially toxic material. Our findings expand understanding of neuronal proteostasis and suggest implications for neurodegenerative disease progression.

Disrupted proteostasis underlies cellular dysfunction in aging and neurodegenerative disease[1], and thus elaborating the cellular mechanisms of protein quality control is a major focus in biomedical research. The cellular commitment to protein quality control engages molecular strategies that include protein refolding via the chaperone network, protein degradation via the ubiquitin proteasome system, and aggregate elimination via autophagy, which work to cull misfolded proteins that otherwise might impair healthy cell functions[2]. Under extreme conditions of proteostress, additional neuroprotective strategies can be engaged. For example, when the proteasome is pharmacologically inactivated, ubiquitinylated proteins are identified by molecular adaptors that link aggregates/misfolded proteins to dynein motors, which in turn transport their cargo along microtubule tracks to deliver the aggregates to a juxtanuclear site called the aggresome[3–5]. The aggresome concentrates and sequesters aggregates within an intermediate filament-surrounded domain; aggresome contents can ultimately be delivered to lysosomes for degradation. Aggregate sequestration by aggresomes is generally considered beneficial since function-disrupting aggregates are prevented from interfering with cytoplasmic activities.

Neurons can engage additional protective strategies under extreme proteostasis disruption. In *C. elegans*, high proteostress that likely overwhelms chaperone, proteasome, and autophagy functions can trigger the production of large (~3.8 μm) vesicles called exophers

[1]Department of Molecular Biology and Biochemistry, Rutgers University, Piscataway, NJ 08855, USA. [2]Department of Neuroscience, Albert Einstein College of Medicine, Rose F. Kennedy Center, Bronx, NY 10461, USA. ✉e-mail: grant@dls.rutgers.edu; driscoll@biology.rutgers.edu

that concentrate, and somewhat selectively remove, aggregated proteins such as transgenically expressed Huntingtin polyQ expansion proteins[6–8]. Extruded neuron-derived exopher vesicles enter the surrounding glial-like hypodermal cell via specialized phagocytosis so that exopher contents are then degraded by the hypodermal cell's lysosomal network[9], neuroprotective[6] transfer biology that may be conserved in flies[10,11] and mammalian neurons[12]. Exopher-like processes have been reported in mammalian systems[13–18], and thus dissection of exophergenesis in the transparent genetic model *C. elegans* may well inform on the fundamental biology of aggregate spreading that occurs in human neurodegenerative disease pathology[12].

Here we report on aggregate accumulation and extrusion in proteostressed *C. elegans* touch receptor neurons. We document roles for conserved adaptor proteins, dynein motors and microtubule integrity in the collection and concentration of aggregating proteins into intermediate-filament-associated compartments that share multiple molecular features of mammalian aggresomes. We identify a cell-autonomous requirement for aggresome-associated intermediate filament (IF) D class proteins IFD-1 and IFD-2, as well as components of the 14-3-3/Hsc70 adaptor complex that can deliver cargo to aggresome-like organelles, in efficient *C. elegans* exophergenesis. Human neuronal intermediate filament hNFL can partially execute touch neuron-autonomous exopher-related functions in *C. elegans* neurons, supporting that IF roles in aggregate transfer during proteostress may be conserved across phyla. Mechanistic linking of conserved aggresome biology with neuronal aggregate extrusion suggests alternative aggregate clearance options for mammalian aggresomes, and invites new considerations for therapeutic design.

## Results

Cellular proteostasis is maintained by balancing protein synthesis and degradation with dedicated removal of misfolded and aggregated proteins via ubiquitin proteosome-mediated degradation and autophagy[2]. Under conditions of extreme proteostress, mammalian cells sequester potentially harmful aggregates in a juxtanuclear aggresome encased by intermediate filament proteins[4,5]. Aggresome contents eventually can undergo lysosomal degradation via autophagy. Details on how aggresomes are eliminated from cells, however, remain a poorly understood facet of proteostasis.

*C. elegans* has served as a highly illuminating model for the elaboration of fundamental mechanisms of aging and proteostasis. Most literature indications of *C. elegans* aggresomes refer to the presence of fluorescently tagged protein aggregates near the nucleus, but little documentation on molecular make up and contents of aggresomes is available in this model[19], especially for neurons. With an interest in the fundamental biology of neuronal proteostasis, we first asked whether aggresomes form in proteo-stressed *C. elegans* neurons. We chose to focus on easily visualized *C. elegans* touch neurons expressing high levels of mCherry (via integrated transgene allele *bzIs166*[P*mec-4*mCherry]), for which previous ultrastructural examination and cell biological characterization provided evidence of high stress and neuronal exopher extrusions[6,7,20]. We refer to this reporter allele as mCherry in the text that follows.

### IFDs colocalize in juxtanuclear foci that are distinct from other organelles

To begin, we sought evidence of aggresome formation by characterizing intermediate filament protein distribution in proteostressed mCherry touch receptor neurons. *ifd-1* and *ifd-2* are documented to be transcriptionally expressed in touch neurons[21–23] and thus we studied the subcellular localization patterns of functional fluorescently tagged GFP::IFD-1 and mNeonGreen::IFD-2 proteins expressed in touch neurons from single copy transgenes. We found that GFP::IFD-1 and mNeonGreen::IFD-2 typically appear in one to three ~1 μm foci located near the nuclear membrane (Fig. 1a, b, see Supplementary Data 1), with

two IFD foci being most commonly observed during early adult life (Supplementary Fig. 1a). The IFD foci in mCherry-stressed neurons increase in size from L4 to adulthood (Fig. 1c, Supplementary Fig. 1b). Additional proteostress, such as overexpression of toxic proteins or genetic impairment of autophagy with the *epg-9(bp390)* mutant, results in a modest increase in the number and size of IFD foci (Supplementary Fig. 1c–e). When animals are stressed with pharmacological proteasome and autophagy inhibitors, foci can increase in size (Supplementary Fig. 1f). IFD foci can be visualized using multiple different fluorescent reporter-tagged versions of IFD-1 and IFD-2 (i.e., for mNeonGreen, RFP, and mScarlet fluorescent tags; Supplementary Fig. 1g–j), indicating that the subcellular localization of IFDs is independent of fluorescent tag identity. Overall, data support that intermediate filaments IFD-1 and IFD-2 concentrate into juxtanuclear foci, the dimensions of which can increase under proteostress.

To determine whether IFD-1 and IFD-2 colocalize with each other at the juxtanuclear foci, we tested for co-incidence in strains that expressed both GFP::IFD-1 and RFP::IFD-2 in touch neurons (Fig. 1d). We find that GFP::IFD-1 and RFP::IFD-2 signals are nearly completely colocalized (quantification in Fig. 1h). To address whether the juxtanuclear IFD foci correspond to known cell organelle compartments, we tracked IFD-1 and IFD-2 localization relative to established Golgi, lysosome, and mitochondrial reporters. We found that tagged IFD proteins did not colocalize substantially with Golgi marker AMAN-2::tagBFP (Fig. 1e, i), late endosome and lysosome marker LMP-1::mScarlet (Fig. 1f, j), or mitochondria tagged with MitoROGFP (Fig. 1g, k). Thus, IFDs localize to a neuronal compartment distinct from Golgi, lysosomes, and mitochondria.

In sum, in proteo-stressed touch neurons, tagged intermediate filaments IFD-1 and IFD-2 concentrate into juxtanuclear foci that have subcellular localization features akin to those described for mammalian aggresomes.

### IF-sized filaments surround aggresome-like structures

We examined ultrastructural features of ALM neuron somata from mCherry proteo-stressed touch receptor neurons for evidence of aggresome-like structures, focusing on 12 distinct ALM touch neurons at adult day 2 (Ad2). mCherry animals were prepared using high pressure freezing/freeze substitution (HPF/FS) and visualized using serial section electron microscopy, and in some cases 3D electron tomography.

We noted that the proteo-stressed ALM somata often contained 1–2 micron-sized rounded structures located close to neuronal nuclei that included a central granular matrix (Supplementary Fig. 2a–d). A striking feature of the juxtanuclear structures was their peripheral association with 10 nm-wide filaments (9.99 nm; > 150 filaments measured, ±0.80 SD), a diameter measure consistent with that of Intermediate Filaments (IF). Electron tomograms confirm that these ~10 nm filaments can be readily distinguished from membrane bilayers (Supplementary Movies 1, 2).

Data from the twelve neurons we examined suggest a potential growth/maturation sequence for the spherical structures. Some of these spherical granular structures contain a patched meshwork-like border of single discontinuous filaments at the periphery, and a few disorganized short loose intermediate filaments inside (Supplementary Fig. 2a, b). Some granular structures that may be more advanced in genesis feature a series of longer curved filaments assembled in parallel at the periphery, with each filament remaining separate from its neighbors, and fewer short filaments inside the structure (Supplementary Fig. 2c). Finally, some rounded juxtanuclear structures display a periphery of loosely-parallel filaments, with some portions of the border having a densely packed layer of filaments with no internal filament structures evident (Supplementary Fig. 2d). We noted that the electron density of the central granular material appears higher in samples with greater filament association. Although technical challenges precluded our immuno-EM verification of intermediate filament

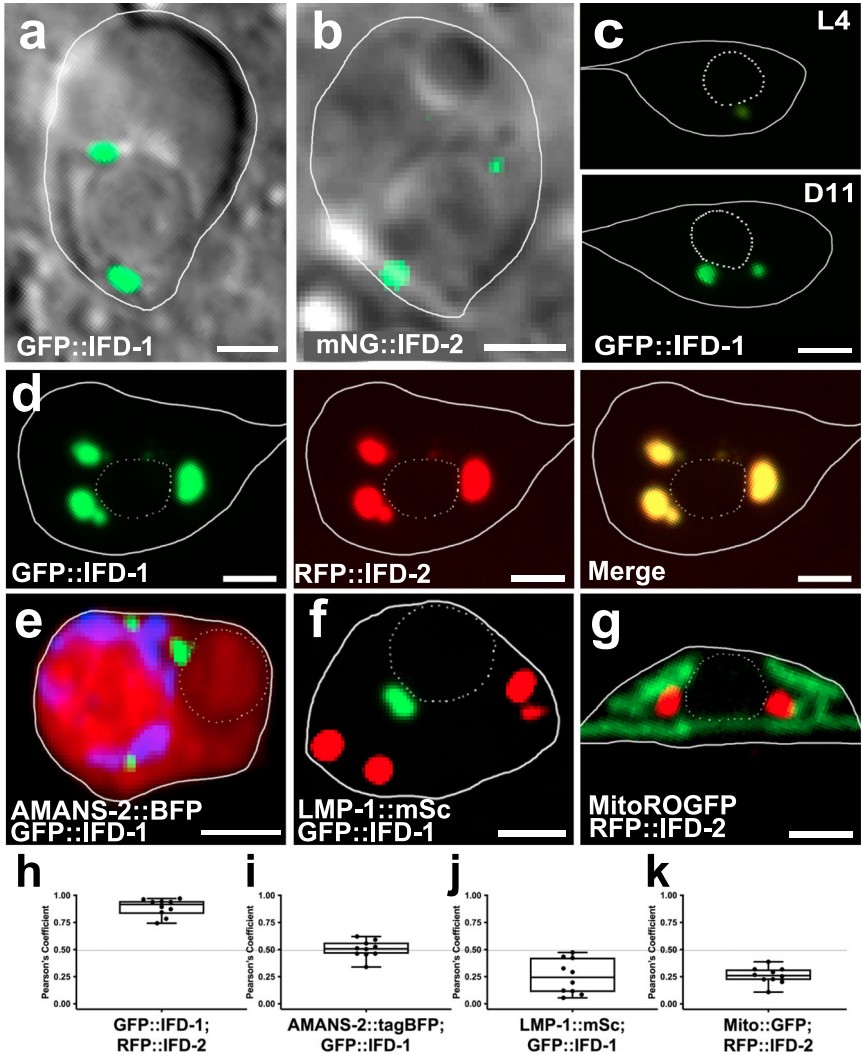

**Fig. 1 | IFD-1 and IFD-2 colocalize to juxtanuclear inclusions that become larger with age and are distinct from other organelles.** Solid white lines outline the touch neuron soma. Scale bar = 2 μm. **a** Ad2 touch neuron, DIC-GFP merge, strain expressing *bzIs166*[P*mec-4*mCherry]; *ifd-1(ok2404)*; *bzSi3*[P*mec-7*GFP::IFD-1]. Image is representative of N > 40 neurons. **b** Ad2 touch neuron, DIC-GFP merge, strain expressing *bzIs166*[P*mec-4*mCherry]; *bzSi37*[P*mec-7*mNeonGreen::IFD-2]. Image is representative of N > 20 neurons with visible mNG expression for this strain. **c** Top: L4 larval stage ALM (Anterior Lateral Microtubule) neuron Bottom: Ad11 ALM neuron. Strain expresses *bzIs166*[P*mec-4*mCherry]; *bzSi3*[P*mec-7*GFP::IFD-1]. Image is representative of N > 50 neurons. **d** Adult touch neuron expressing *bzEx279*[P*mec-7* GFP::IFD-1 P*mec-7*RFP::IFD-2]. Image is representative of N > 20 neurons. **e** Adult touch neuron expressing *bzIs166*[P*mec-4*mCherry]; *bzEx265*[P*mec-4*TagBFP::AMAN-2]; *bzSi3*[P*mec-7*GFP::IFD-1]. Image is representative of N > 20 neurons. **f** Adult touch neuron expressing *bzIs3*[P*mec-7*GFP::IFD-1]; *pwSi222*[P*mec-7*LMP-1::mScarlet]. Image is representative of N > 20 neurons. **g** Adult touch neuron expressing

*bzIs166*[P*mec-4*mCherry]; *bzEx265*[P*mec-4*TagBFP::AMAN-2]; *bzSi3*[P*mec-7*GFP::IFD-1]. Image is representative of N > 20 neurons. **h** Colocalization correlation of GFP::IFD-1 and RFP::IFD-2 signals (**d**) graphed as Pearson's Coefficient of red and green channel; boxes indicate the coefficient range with a minimum (0.74), maximum (0.97), and mean (0.89); N = 10 neurons. **i** Colocalization correlation of GFP::IFD-1 and TagBFP::AMAN-2 signals (**e**) graphed as Pearson's Coefficient of blue and green channel; boxes indicate the coefficient range, with a minimum (0.34), maximum (0.62), and mean (0.50); N = 10 neurons. **j** Colocalization correlation of GFP::IFD-1 and LMP-1::mSc signals (**f**) graphed as Pearson's Coefficient of red and green channel; boxes indicate the coefficient range, with a minimum (0.05), maximum (0.47), and mean is (0.42); N = 10 neurons. **k** Colocalization correlation of RFP::IFD-2 and mitoROGFP (**g**) graphed as Pearson's Coefficient of red and green channel; boxes indicate the coefficient range, with a minimum (0.11), maximum (0.39), and mean (0.26); N = 10 neurons.

identity, the structures we documented in mCherry-expressing ALM neurons are strikingly reminiscent of maturing mammalian aggresomes that form under high proteostress and are characterized by intermediate filament proteins that coalesce around sequestered aggregated protein[3,5,24,25], consistent with our florescence microscopy visualization of tagged IFD proteins in proteo-stressed neurons.

## IFD foci depend on MTs and dynein motors for formation

Under high proteostress conditions, ubiquitinated and non-ubiquitinated aggregated proteins in mammalian (and other) cells can be linked to adaptors such as HDA-6/HDAC6, SQST-1/p62, and

the HSP-1/Hsc70/BAG3/FTT-2/14-3-3 complex, which connect to dynein motors and move along microtubule tracks to juxtanuclear sites to deposit aggregate cargos[26–28] (summary diagram in Fig. 2a). IFs concentrate at these juxtanuclear aggresome-like organelle sites. To address whether formation of the *C. elegans* IF compartment features reflects aggresome-like biology, we perturbed key aspects of aggresome biogenesis and probed for colocalization of IFDs and homologs of mammalian aggresome proteins.

We tested for microtubule and dynein roles in IFD-1 puncta formation (Fig. 2b, c). To test microtubule (MT) requirements for juxtanuclear IFD concentration, we treated P*mec-7*GFP::IFD-1 expressing animals with the potent microtubule inhibitor colchicine[29] from

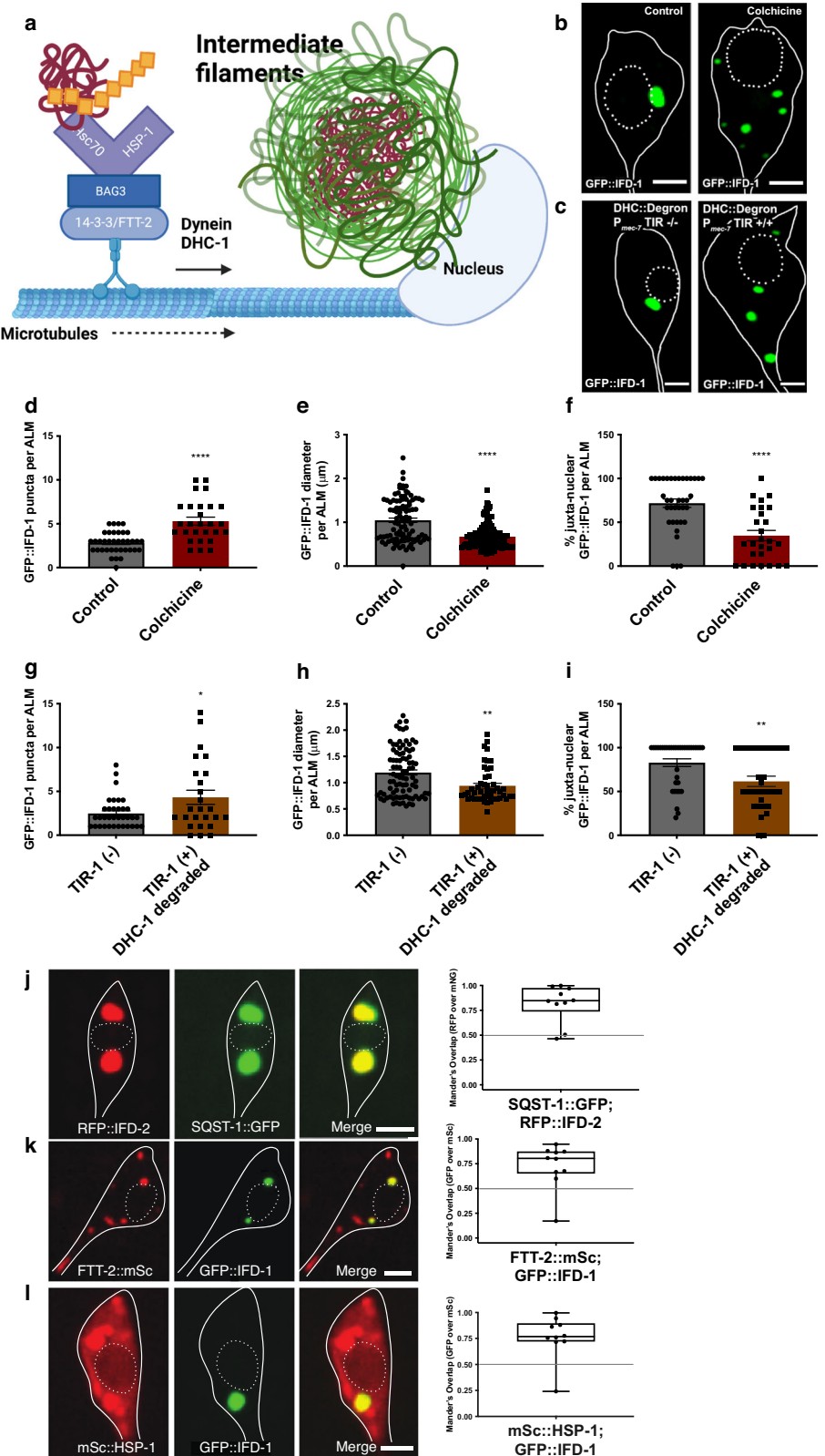

L4 to Ad2 (Fig. 2b). We find that colchicine increases the number of dispersed GFP::IFD-1 foci (Fig. 2d), reduces their size (Fig. 2e), and disrupts their juxtanuclear positioning (Fig. 2f), suggesting a requirement for MTs in the development and juxtanuclear localization of the *C. elegans* IFD compartment, as occurs for mammalian aggresomes.

Colchicine treatment could act via non-autonomous influences. To address cell-intrinsic requirements, we used the auxin-inducible degradation system[30] to degrade degron-tagged dynein heavy chain DHC-1 specifically in the touch neurons. We found that in the presence of auxin (i.e., when DHC-1::degron protein was depleted), IFD inclusions were more numerous (Fig. 2c; Fig. 2g), smaller (Fig. 2h), mobile,

**Fig. 2 | Genesis of juxtanuclear IFD inclusions depends on functional microtubules and dynein; aggregate adaptor proteins FTT-2/14-3-3 and HSP-1/Hsc70 colocalize with IFD-positive organelles. a** 14-3-3(FTT-2) associates with BAG3-bound Hsc70/(HSP1); Hsc70(HSP-1) recognizes and binds ubiquitinylated aggregates, which are transported via microtubules to the aggresome-like compartment. *C. elegans* gene names are indicated. **b-c**, **j-l** Solid white lines outline the soma cell body; white dashed lines outline the nucleus; scale bar = 2 μm. **b** Touch neuron expressing *bzIs166*[P*mec-4*mCherry]; *bzSi3*[P*mec-7*GFP::IFD-1]. Left: Control Ad2 DMSO; Right: Ad2 ALM, 5 mM colchicine in DMSO added from L4 to Ad2. Images are representative of N > 90 ALMs. **c** AID dynein heavy chain knockdown strain expresses *bzIs166*[P*mec-4*mCherry]; *bzSi6*[P*mec-7*TIR1::TagBFP]; *bzSi3*[P*mec-7*GFP::IFD-1]; *dhc-1(ie28*[DHC-1::degron::GFP]); control strain lacks TIR1; 5 mM auxin exposure from L4 to Ad2. Images are representative of N > 90 ALMs. **d-i**. Data are 3 trials, two-tailed *t* test, error bars SEM. **d** Number of IFD-1-positive puncta per Ad2 ALM in *bzIs166*[P*mec-4*mCherry]; *bzSi3*[P*mec-7*GFP::IFD-1]. Control DMSO; colchicine 5 mM, L4 to Ad2 exposure. *N* = 37 and 27 ALM, respectively; ****P = 0.0001. **e** Diameter (μm) of IFD-1-positive foci in Ad2 ALM. Strain and treatment are the same as in (**b**). Ad2 N = 101 and 143 ALM respectively; ****P = 0.0001. **f** Percentage of juxtanuclear foci. Strain and treatment are the same as in (**b**). *N* = 37 and 27 ALM respectively, ****P = 0.0001. **g** Number of IFD-1-positive puncta per Ad2 ALM. Strains are

*bzIs166*[P*mec-4*mCherry]; *bzSi3*[P*mec-7*GFP::IFD-1]; *dhc-1(ie28*[DHC-1::degron::GFP]) and *bzIs166*[P*mec-4*mCherry]; *bzSi6*[P*mec-7*TIR1::TagBFP]; *bzSi3*[P*mec-7*GFP::IFD-1]; *dhc-1(ie28*[DHC-1::degron::GFP]); auxin exposure from L4 to Ad2. N = 35 and 25 ALM respectively, *P = 0.019. **h** Diameter (μm) of IFD-1-positive foci in Ad2 ALM. Strains are the same as in (**c**), with control −/− lacking P*mec-7*TIR1 and both strains exposed to 5 mM auxin from L4 to Ad2. N = 98 ALM and 48 ALM respectively, **P = 0.0011. **i** Percentage of juxtanuclear foci for strains presented in (**c**). *N* = 36 ALM and 31 ALM respectively, **P = 0.0041. **j-l** Boxes indicate the range of coefficient. **j** Representative of adult touch neuron expressing *bpIs151*[P*sqst-1*SQST-1::GFP+ *unc-76*(+)]; *bzEx261*[P*mec-7*RFP::IFD-2]. Quantification of colocalization of *N* = 10 ALMs, graphed on the right as Mander's overlap coefficient of mScarlet channel over GFP channel. Range minimum (0.46), maximum (0.99), and mean (0.82) coefficient. **k** Representative of adult touch neuron expressing *bzSi51*[P*mec-7*FTT-2::mSc]; *bzSi3*[P*mec-7*GFP::IFD-1]. Quantification of colocalization of *N* = 10 ALMs, graphed on the right as Mander's overlap coefficient of GFP channel over mScarlet channel. Range minimum (0.17), maximum (0.95), and mean (0.73) coefficient. **l** Representative of adult touch neuron expressing *bzSi53*[P*mec-7*mSc::HSP-1]; *bzSi3*[P*mec-7*GFP::IFD-1]. Quantification of colocalization of *N* = 10 ALMs, graphed as Mander's overlap coefficient of GFP channel over mScarlet channel. Range minimum (0.24), maximum (0.99), and mean (0.77) coefficient.

and more dispersed (Fig. 2i) as compared to non-knockdown DHC-1(+) controls. We conclude that a DHC-1-containing dynein motor acts cell autonomously in IFD collection and concentration into large juxtanuclear inclusions, similar to dynein motor roles in mammalian aggresome formation.

### IFDs associate with adaptor proteins SQST-1, HSP-1 and FTT-2/14-3-3

We also tested for molecular associations of IFDs with homologs of aggregate adaptor proteins documented to contribute to, and associate with, mammalian aggresomes[3,28,31,32], including the SQSTM1/p62 autophagy adaptor, HDAC6 deacetylase; and 14-3-3 protein which functions in an adaptor complex that includes BAG3 and chaperone Hsc70[32] (see Fig. 2a). We constructed strains harboring fluorescently tagged IFD-1 and *C. elegans* homologs of *sqst-1*/SQSTM1, *hsp-1*/Hsc70 and *ftt-2*/14-3-3. For each adaptor tested, we find colocalization with the juxtanuclear IFD-1 protein, akin to its characterized mammalian counterpart (Fig. 2j–l).

To test the possibility that aggresome adaptors are needed to form IFD-aggresome-like structures, we examined available viable mutants for *C. elegans* counterparts of three aggregate collection pathways: *hda-6*/HDAC6, *sqst-1*/SQSTM1, and *ftt-2*/14-3-3. We scored aggresome-like organelle dimensions as reported by GFP::IFD-1 in adaptor mutant backgrounds (Supplementary Fig. 3). Notably, we find that no individual adaptor disruption eliminates the IFD compartment. Thus, no single aggregate collection pathway appears essential for aggresome-like organelle formation, and multiple routes likely function in parallel for cargo delivery, consistent with adaptor requirements reported for mammalian models. We did quantitate some changes in aggresome-like organelle dimensions in *sqst-1(ok2892)*, in which ALM touch neurons have significantly smaller GFP::IFD-1 puncta (Supplementary Fig. 3a), and increased puncta numbers (Supplementary Fig. 3b). In addition, Ad1 *ftt-2(n4426)* ALM somata have significantly smaller GFP::IFD-1 aggresome-like organelles without a change in number of aggresome-like organelles per neuron (Supplementary Fig. 3c, d). *hda-6*/HDAC6 deletion did not change size or number of GFP::IFD-1 puncta (Supplementary Fig. 3e, f). Our data are consistent with roles for SQST-1 and FTT-2 in aggresome-like organelle filling.

### Aggresome-like organelles include ubiquitin and neurotoxic HttPolyQ

Mammalian aggresomes can collect ubiquitinated proteins[3] and therefore we also asked whether IFD inclusions colocalize with sites of ubiquitin concentration. We co-expressed single copy transgene

mNeonGreen::UBQ-2 in touch neurons together with mScarlet::IFD-1 (Fig. 3a) or mScarlet::IFD-2 (Fig. 3b) and found that although some fluorescently tagged ubiquitin is dispersed or concentrated elsewhere in the neuronal soma as expected, mNG-tagged ubiquitin also concentrates in the IFD-1 and IFD-2 inclusions. We conclude that like mammalian aggresomes, the *C. elegans* neuronal juxtanuclear IFD inclusions are associated with concentrated ubiquitinylated proteins. Together, our data highlight homology of biogenesis and composition between *C. elegans* neuronal IFD structures and mammalian aggresomes.

Ubiquitin colocalization (Fig. 3a, b) is suggestive of concentrated misfolded/aggregated proteins. To address whether a characterized toxic aggregating human neurodegenerative disease protein can concentrate in the IFD-aggresome-like inclusions, we focused on polyglutamine expansion proteins, which aggregate in *C. elegans* neurons[33,34]. We constructed and co-expressed a single copy transgene encoding mNG::HttQ74 (first exon of the human Huntingtin gene fused to a HttQ74 expansion), together with mScarlet::IFD-1 or mScarlet::IFD-2 reporters. We identified strong localization of mNG::HttQ74 to IFD-1 and IFD-2 inclusions in the touch neuron soma when mNG::HttQ74 is expressed at levels at which visible aggregation is present (Fig. 3c, d). We also examined a HttQ128::CFP line expressed from an integrated high-copy-number array along with mScarlet::IFD-1 to record coincident fluorescence (Fig. 3e). Our data establish that expanded polyglutamine proteins can concentrate at IFD inclusions, consistent with the idea that IFDs mark and sequester a domain of aggregated proteins, similar to what has been reported for mammalian aggresomes.

Interestingly, we noticed that IFDs and HttQ128::CFP can adopt multiple relative configurations, with IFDs and polyQ aggregates being mostly coincident for smaller (presumptively newly formed) aggresome-like organelles in early adult life (Fig. 3c–e), but more frequently with HttQ128::CFP exhibiting less complete IF covering with advancing time or when relative expression of HttQ128::CFP and IFDs appears unbalanced, with more HttQ128::CFP evident (Fig. 3f, Supplementary Fig. 4). Whether these structures correspond to IF domains overrun by high aggregate concentrations, or aggresomes dissociating for degradation remains for future testing.

We also examined localization of the highly expressed mCherry reporter relative to IFD foci, which revealed a different pattern. Our previous work showed that mCherry *bzIs166*[P*mec-4*mCherry] expression increases proteostress and that mCherry can concentrate into subcellular foci, although juxtanuclear positioning was not a hallmark of cellular positioning[6,7]. We found dimmer mCherry concentrations colocalized with IFD-1 in some ALM neurons, but most highly

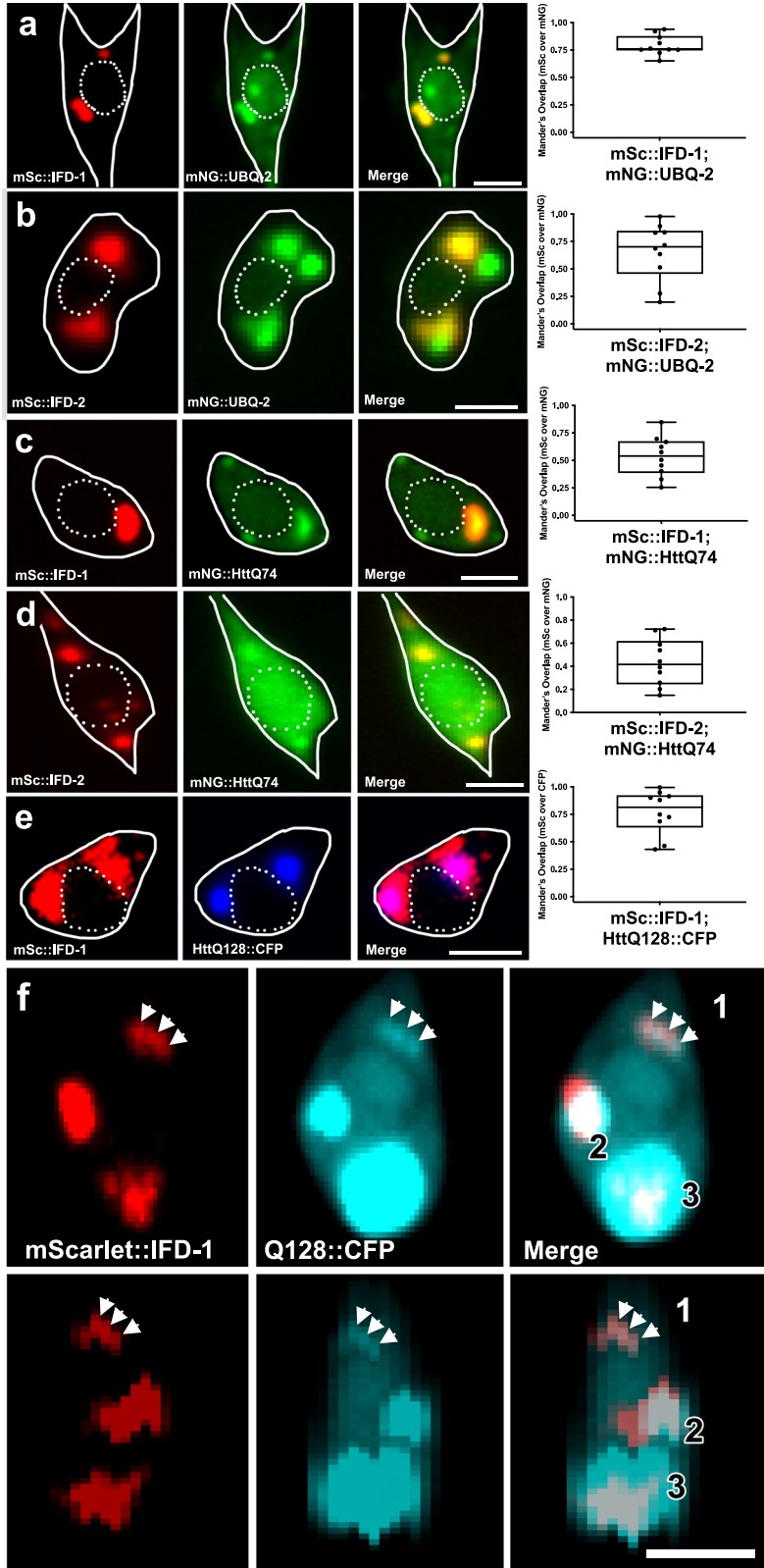

concentrated mCherry in touch neuron somata was within LMP-1::GFP positive lysosomes (LMP-1 is the lysosomal membrane protein LAMP1 ortholog[35]; Supplementary Fig. 5a, b).

In summary, colocalization analyses indicate that proteins of different character (e.g., expanded polyQ proteins and mCherry) can be differentially handled, concentrated, and/or stored in proteostressed neurons (Supplementary Fig. 5), and that the expanded polyQ proteins associate with, and linger within, IFD inclusions in proteostressed neurons (Fig. 3, Supplementary Fig. 4).

## IFDs are required for efficient neuronal exopher production

In human neurodegenerative disease, aggregates can transfer among neurons and glia to promote pathological spread[12]. The extent to which mammalian aggresome pathways contribute to spreading

**Fig. 3 | Disease-associated human Htt-polyglutamine expansion protein dynamically colocalizes with IFD-1 in touch neurons. a–e** White solid outline indicates the soma cell body and white dashed line outlines the nucleus; scale bar = 2 μm. Colocalization of $N = 10$ ALMs is graphed as Mander's overlap coefficient of mScarlet channel over mNeonGreen channel, with boxes indicating the coefficient range. **a** Representative adult touch neuron expressing single copy *bzSi38*[P*mec-7*mNG::UBQ-2]; *bzSi34*[P*mec-7*mSc::IFD-1]; range minimum (0.65), maximum (0.94), and mean (0.79) coefficient. **b** Representative adult touch neuron expressing *bzEx422*[P*mec-7*mNG::UBQ-2]; *bzEx421*[P*mec-7*mSc::IFD-2]; range minimum (0.20), maximum (0.98), and mean (0.66) coefficient. **c** Representative adult touch neuron expressing *bzSi39*[P*mec-7*HttQ74::mNG]; *bzSi34*[P*mec-7*mSc::IFD-1]; range minimum (0.26), maximum (0.85), and mean (0.59) coefficient. Images selected were those in which mNG was visibly aggregated in the cell, as opposed to homogeneous dim cytosolic signal. **d** Representative adult touch neuron

expressing b*zSi40*[P*mec-7*mNG::HttQ74]; *bzEx420*[P*mec-7*mSc::IFD-2] that displays visible mNG aggregation; range minimum (0.15), maximum (0.72), and mean (0.57) coefficient. **e** Representative adult touch neuron expressing *igIs1*[P*mec-7*YFP P*mec-3*HttQ128::CFP]; *bzSi34*[P*mec-7*mSc::IFD-1] that displays visible CFP and mSc co-expression; range minimum (0.43), maximum (0.99), and mean (0.56) coefficient. **f** HttQ128::CFP is expressed from a high-copy-number integrated transgene array. The IFD compartment often does not appear to fully surround the HttQ128 signal. Top row: Adult touch neuron soma displaying the three main patterns of interaction we note in lines that express mScarlet::IFD-1 and HttQ128::CFP. (1) HttQ128 and IFD overlap entirely (white arrows). (2) HttQ128 and IFD display a bi-lobed globular interaction, where part of the bi-lobed structure has overlap of HttQ128 and IFD. (3) HttQ128::CFP signal fully encompasses the IFD signal. Bottom row: side view z-stack (0.2 μm) from the image in the top row.

mechanisms has not been investigated. Our mCherry model is well suited to address this question, as we previously showed that adult neurons in this background can selectively eject material in large (~3.8 μm) membrane-bound vesicles called exophers that can include protein aggregates[6]. Exopher production begins with outward budding of a nearly soma-sized membrane-bound vesicle that can selectively include damaged organelles and toxic proteins[6,7] (Fig. 4a, b); outward budding of the large membrane-surrounded vesicle delivers exopher contents directly to neighboring tissues (the glia-like hypodermis, in the case of *C. elegans* touch-neuron-derived exophers) for degradation[6,9]. We have speculated that the *C. elegans* exopher mechanism may be analogous to the poorly understood process by which neurodegenerative disease-associated aggregates spread from cell to cell to promote pathology in human neurodegenerative disease.

Having shown that *C. elegans* neuronal aggresome-like organelles can store/associate with neurotoxic disease aggregates, we considered their potential roles in aggregate transfer biology. Potential intermediate filament requirements in exophergenesis were an initial focus given that in humans, circulating IFs are clinical biomarkers of human neurodegenerative disease[36], and IFs are major components of α-synuclein-containing Lewy bodies that characterize Parkinson's disease neuropathology[37,38], features that might intersect with extrusion biology and proteostasis balance.

We tested whether null alleles of *ifd-1* and *ifd-2* act as exopher modulators. We obtained a mutant strain containing the *ifd-1(ok2404)* deletion (1863 base pairs deleted) and used CRISPR/Cas9 technology to generate deletion allele *ifd-2(bz187)*, which is missing the *ifd-2* start codon and the first three *ifd-2* exons are affected. The *ifd-1(Δ)* and *ifd-2(Δ)* mutants exhibited WT morphology (Supplementary Fig. 6a, b) and brood size (Supplementary Fig. 6c), as well as normal or near normal developmental timing (Supplementary Fig. 6d, e) *(ifd-2(Δ)* ~ 5 h average delay to L4) and lifespan (Supplementary Fig. 6f, g)). *ifd-1(Δ)* and *ifd-2(Δ)* mutants also exhibited WT osmotic stress survival (Supplementary Fig. 6h), normal *(ifd-2(Δ))* or elevated oxidative stress sensitivity (for *ifd-1(Δ)*) (Supplementary Fig. 6i; osmotic and oxidative stresses are associated with proteostasis disruption), and WT or modest suppression *(ifd-2(Δ))* of baseline *gst-4*::GFP expression (GST-4::GFP is a *daf-16*/FOXO/ SKN-1/NRF responsive glutathione-S-transferase reporter commonly used to report stress conditions[39]) (Supplementary Fig. 6j, k). In terms of touch neuron-specific features, we recorded normal mechanosensory response (Supplementary Fig. 7a) and touch neuron morphology during aging (Supplementary Fig. 7b). Likewise, in *ifd* null mutants, mCherry distribution and levels are not markedly changed (Supplementary Fig. 7c, d). We conclude that *ifd-1* and *ifd-2* deletions do not confer substantial systemic physiological disruption in the whole animal; nor do *ifd-1* and *ifd-2* deletions markedly perturb the basic biology of the touch neurons in which we characterize exophergenesis.

We measured exophers at Ad2 in the mCherry; *ifd-1(Δ)* and *ifd-2(Δ)* backgrounds to find that *ifd-1(Δ)* and *ifd-2(Δ)* each suppress exopher

production (Fig. 4c), establishing an impact of *C. elegans* IFDs in exophergenesis. We confirmed exopher suppression in *ifd-1(Δ)* and *ifd-2(Δ)* using alternative reporter cargo encoded by *uIs31*[P*mec-17*GFP] (Supplementary Fig. 8), and consequent to *ifd-1* and *ifd-2* RNAi knockdown in a pan-neuronal specific RNAi knockdown strain (Supplementary Fig. 9a). Combined data define a role for *C. elegans* IFDs in promoting exopher levels in proteostressed neurons.

## IFD-1 and IFD-2 act in the same pathway to promote exopher formation

Despite the fact that null alleles of *ifd-1* and *ifd-2* remove extensive protein coding sequences, neither *ifd-1(Δ)* nor *ifd-2(Δ)* fully eliminated exophers. To address whether *ifd-1* and *ifd-2* might act in the same, or in distinct, pathways in exophergenesis, we constructed an *ifd-1(Δ); ifd-2(Δ)* double mutant and compared exopher levels to those in single *ifd* mutants (Fig. 4c). We found that exopher production in the *ifd-1(Δ); ifd-2(Δ)* double mutant was suppressed to the same extent as in the single mutants, suggesting that *ifd-1* and *ifd-2* are likely to act in the same pathway to influence exopher production, and consistent with their colocalization in neurons (Fig. 1d).

Some exopher production remains evident when both *ifd-1* and *ifd-2* are absent, suggesting that other IF proteins might also influence exopher production. Indeed, when both IFDs are absent, RNAi knockdown of other *C. elegans* IF genes in a strain sensitized for touch neuron RNAi suggests that additional IF proteins may modulate TN exophergenesis (Supplementary Fig. 9b). Given the relatively strong exopher suppression effects for *ifd-1(Δ)* and *ifd-2(Δ)*, the RNA-seq-documented *ifd* expression in touch neurons[21–23], our initial extensive characterization of the fluorescently tagged IFD-1 and IFD-2 in touch neurons (Fig. 1, Supplementary Fig. 1), and the limited study of the biological roles of the *C. elegans* IFD genes to date[40–44], we elected to focus on *ifd-1* and *ifd-2* contributions to exophergenesis in further studies we present here.

## IFDs act cell autonomously in neurons to promote exopher production

To confirm that *ifd* mutations themselves, rather than unintended mutations present in the strain background, modulate exopher levels, we complemented *ifd(Δ)s* with cognate *ifd* transgenes. We used a single-copy GFP::IFD-2 transgene, expressed from the native *ifd-2* promoter, and tested for rescue of the *ifd-2(Δ)* phenotype. *bzSi76*[P*ifd-2*GFP::IFD-2] rescues the *ifd-2(Δ)* exopher deficit, confirming the *ifd-2* gene role in modulating exopher levels (Fig. 4d). We relied on single copy touch neuron-specific expression of GFP::*ifd-1* crossed to *ifd-1(Δ)* to confirm rescue of the exopher phenotype (Fig. 4e).

To address the question of whether *ifd-1* and *ifd-2* exert cell-autonomous roles for IFDs in neuronal exopher production, we expressed single-copy fluorescently-tagged reporters exclusively in the touch neurons under the control of the *mec-7* touch neuron

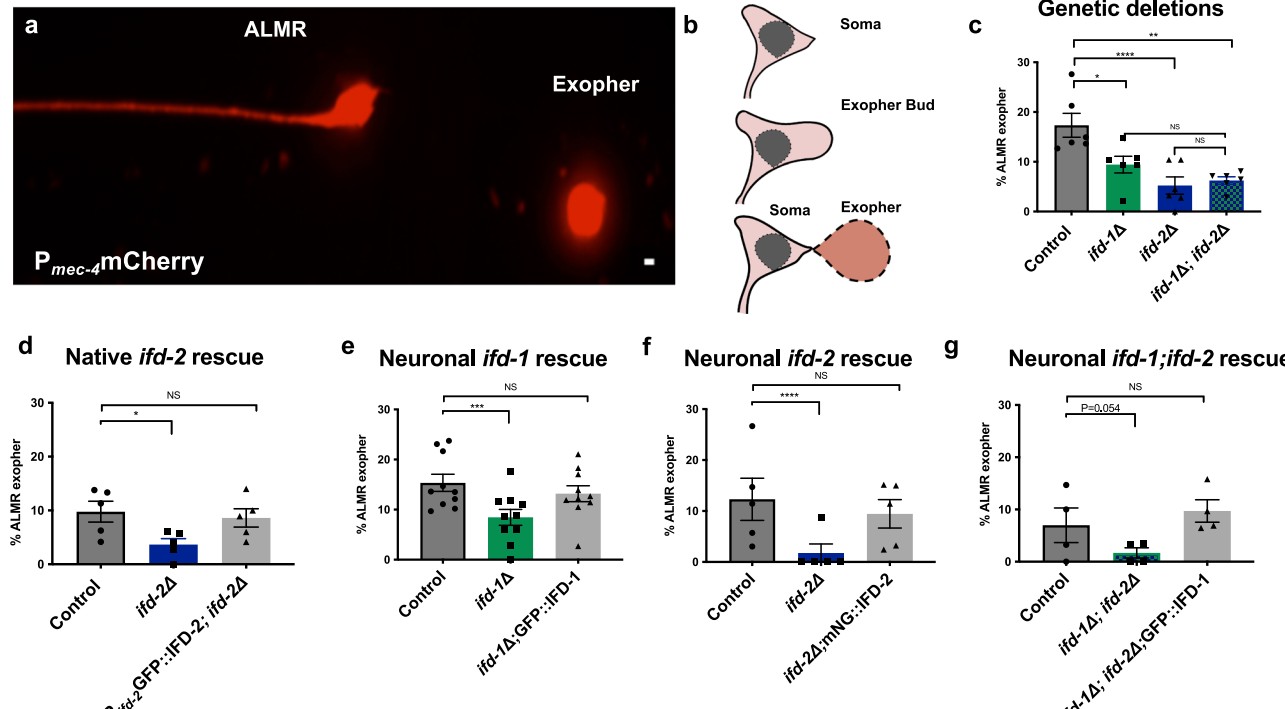

**Fig. 4 | Intermediate filaments IFD-1 and IFD-2 act touch-neuron autonomously to support exopher production. a** A typical exopher is readily visualized by mCherry in strain expressing *bzIs166*[P*mec-4*mCherry]. mCherry is a fluorophore expressed from an integrated P*mec-4*-driven high copy number transgene array; high expression mildly induces exophergenesis. ALMR (Anterior Lateral Microtubule, Right side) touch neuron and an ALMR-derived exopher are pictured. Scale bar = 2 μm. Image is representative of *N* > 1000 exophers. **b** Cartoon depiction of the exophergenesis process. (Top) A normal neuronal soma. (Middle) The neuronal soma swells, cellular contents can polarize, and a swelling and budding-out process generates a prominent exopher bud. (Bottom) The exopher bud continues to extend outward; the exopher bud matures into a distinct exopher. **c–g** Exopher studies were analyzed using Cochran-Mantel-Haenszel (CMH) test. **c** We scored ALMR exopher production on Ad2 in the *ifd-1(ok2404)* deletion mutant, the *ifd-2(bz187)* deletion mutant, and the double *ifd* deletion mutant *ifd-1(Δ); ifd-2(Δ)*, all of which harbored the exopher-inducing *bzIs166*[P*mec-4*mCherry] transgene. *N* = 389, 438, 422, and 401 ALMR for control and mutants respectively. The ALMR exopher percentage in the *ifd-1(Δ); ifd-2(Δ)* double mutant is not significantly different from the single mutant *ifd-1* (*P* = 0.351); double mutant compared to *ifd-2* (*P* = 0.0952). 6 trials. **d** We expressed GFP::IFD-2 from the native *ifd-2* promoter in the mCherry

background and scored for exopher levels. We compared ALMR exophers at Ad2 in *bzIs166*[P*mec-4*mCherry] to *bzIs166*[P*mec-4*mCherry]; *ifd-2(bz187)* (*P* = 0.017). *bzIs166*[P*mec-4*mCherry] compared to *bzIs166*[P*mec-4*mCherry]; *ifd-2(bz187); bzSi76*[P*ifd-2*GFP::IFD-2] is not significantly different (*P* = NS). *N* = 239, 240, and 230 ALMR for control, mutant, and rescue respectively. 5 trials. **e** We scored ALMR exophers at Ad2 in *bzIs166*[P*mec-4*mCherry] compared to *bzIs166*[P*mec-4*mCherry]; *ifd-1(ok2404)*. (***P* = 0.00047). P*mec-4*mCherry is not significantly different compared to *bzIs166*[P*mec-4*mCherry]; *ifd-1(ok2404); bzSi3*[P*mec-7*GFP::IFD-1] (*P* = 0.43). *N* = 918, 549, and 555 ALMR for control, mutant, and rescue respectively. 10 trials. **f** We scored ALMR exophers at Ad2 in *bzIs166*[P*mec-4*mCherry] and compared to *bzIs166*[P*mec-4*mCherry]; *ifd-2(bz187)* (****P* = 0.000067). *bzIs166*[P*mec-4*mCherry] is not significantly different from *bzIs166*[P*mec-4*mCherry];*ifd-2(bz187); bzSi37*[P*mec-7*mNeonGreen::IFD-2] (*P* = 0.40). *N* = 182, 179, and 180 ALMR for control, mutant, and rescue respectively. 5 trials. **g** We scored ALMR exophers at Ad2 in *bzIs166*[P*mec-4*mCherry] compared to *bzIs166*[P*mec-4*mCherry]; *ifd-1(ok2404); ifd-2(bz187)* (*P* = 0.054). *bzIs166*[P*mec-4*mCherry] compared to *bzIs166*[P*mec-4*mCherry]; *ifd-1(ok2404); ifd-2(bz187); bzEx270*[P*mec-7* GFP::IFD-1 OE] is NS. *N* = 127, 136, and 129 ALMR in control, mutant, and rescue respectively, 4 trials.

promoter. We found that single copy *bzSi3*[P*mec-7*GFP::IFD-1] rescues the exopher deficit phenotype of the *ifd-1(Δ)* mutant (Fig. 4e). Similarly, single copy *bzSi37*[P*mec-7*mNeonGreen::IFD-2] rescues the exopher deficit phenotype of the *ifd-2(Δ)* mutant (Fig. 4f). Touch-neuron-specific genetic complementation supports that IFDs are required cell-autonomously for neuronal exophergenesis.

Because *ifd-2* is also expressed in the intestine[42,43] and *ifd-2* mutants can have intestinal morphology defects[43], we also tested for rescue of the neuronal exopher phenotype when we expressed single copy mNeonGreen::IFD-2 only in the intestine (*bzSi45*[P*vha-6*mNeonGreen::IFD-2]). Comparing *ifd-2(Δ)* to *ifd-2(Δ)*; P*vha-6*mNeonGreen::IFD-2, intestinal expression of mNeonGreen::IFD-2 does not rescue the neuronal exopher phenotype of *ifd-2(Δ)* (Supplementary Fig. 10), consistent with a predominant role for *ifd-2* in exophergenesis in the neuron (Fig. 4e–g). A trend toward partial rescue, however, leaves open the possibility of some intestinal contribution. Overall, touch neuron-specific rescue of the exopher production deficits in *ifd* null mutants support that IFD-1 and IFD-2 primarily act

autonomously within the touch neuron to influence exopher production.

**Over-expression of *ifd-1* or *ifd-2* does not change exophergenesis**
Our gene manipulations of *ifd-1* and *ifd-2* also enabled us to address consequences of high level expression of *ifd-1* and *ifd-2* on exopher levels (Supplementary Fig. 11, Supplementary Fig. 12). In brief, over-expression of functional *ifd-1* or *ifd-2* constructs in an *ifd*(+) background does not change exopher levels (Supplementary Fig. 11). Lack of over-expression effects (Supplementary Fig. 11) are consistent with our observation that size of the aggresome-like compartment per se does not correlate with high exopher levels and aggresome-like organelle size does not predict extrusion (Supplementary Fig. 13).

We find some capacity for cross-complementation by high copy *ifd* expression (i.e., *ifd-1(Δ)* by *ifd-2*OE; Supplementary Fig. 12a). Thus, despite the clear genetic requirements for native levels of both *ifd-1* and *ifd-2* for normal exopher production levels (Fig. 4c, Supplementary Fig. 8b), the partial functional substitution conferred by cross-

overexpression supports similar bioactivity for *ifd-1* and *ifd-2* (Fig. 4g, Supplementary Fig. 12).

## Adaptor proteins SQST-1, HDA-6, and FTT-2 modulate exopher production

Having established that *ifd-1* and *ifd-2* can modulate exopher levels, we sought additional evidence of aggresome component impact on exophergenesis. We scored exopher levels in available mutants for genes encoding aggresome adaptor proteins that function in aggresome loading of cargo and colocalize with IF-organelles (Fig. 2j–l, Supplementary Fig. 3), appreciating at the outset that the genes we tested encode multi-tasking proteins that contribute in multiple facets of proteostasis.

We found that *sqst-1(ok2892)* increases variation in exopher levels (Supplementary Fig. 14a), without conferring a statistically significant change. *hda-6(ok3203)* (which did not significantly impact GFP::IFD-1 formation (Supplementary Fig. 3e, f)), increased exopher levels (Supplementary Fig. 14b).

We found that the 14-3-3 mutant *ftt-2(n4426)* exhibits a clear exopher suppressor phenotype on Ad1, implicating FTT-2 in exopher production (Supplementary Fig. 14c). To begin to address cell autonomy of FTT-2/14-3-3 for touch neuron exopher production, we induced *ftt-2*(RNAi) knockdown in both neuron-specific and touch neuron-sensitized RNAi backgrounds. Our data reveal a suppression of exopher formation when *ftt-2* is targeted for knockdown in neurons (Supplementary Fig. 14d, e). We also introduced a single copy of *ftt-2* expressed from the touch neuron *mec-7* promoter (P$_{mec-7}$FTT-2::mSc; P$_{mec-4}$mCherry) in an otherwise WT *ftt-2(+)* background. When we tested exopher levels in this strain, which is wild type *ftt-2* at the endogenous genomic locus and harbors a second copy of P$_{mec-7}$FTT-2::mSc inserted via MiniMos single site insertion[45] (*i.e.*, at least 2X *ftt-2* gene dose), we quantitated significantly increased TN exopher levels (Supplementary Fig. 14f). Data support cell autonomous activity of FTT-2 in touch neuron exopher modulation and suggest that enhanced FTT-2 activity can drive exopher formation.

To assess requirements for the Hsc70/BAG3/14-3-3 adaptor complex in exopher biology from a second angle, we knocked down *hsp-1/Hsc70*[26,32] in a touch neuron-RNAi sensitized background (Supplementary Fig. 14g, h). Exopher production scores suggest that the FTT-2 aggresome-related binding partner HSP-1 plays a role in touch neuron exophergenesis and further support that an FTT-2/HSP-1 complex can act to modulate exopher levels.

In sum, genetic perturbations of IF genes *ifd-1* and *ifd-2*, and aggresome adaptor genes *ftt-2* and *hsp-1* suppress exopher levels, demonstrating a potential link between these aggresome components and exopher formation.

## Aggresome-like foci can be expelled within exophers

Having established that specific aggresome components likely act cell autonomously to influence exopher levels, we asked whether aggresome-like inclusions can themselves be expelled in exophers, and if so, whether aggresome inclusion is mandatory for exopher formation. We first examined exophers produced by neurons co-expressing mCherry and GFP::IFD-1 to find that exophers can contain extruded GFP::IFD-1 foci in this background (Fig. 5a; GFP::IFD-1 foci in 11/61 exopher events, Fig. 5b). With regard to the capacity to fully clear neurons of the aggresome-like structures, in a separate experiment, we find that 4/9 exopher events that include aggresome-like foci, the complete visualizable GFP::IFD-1 aggresome-like organelle contents of the soma were ejected; in the remaining 5 cases, one GFP::IFD-1-aggresome-like structure was retained in the soma and one was extruded in the exopher (Fig. 5c).

Since IFDs colocalize with HttPolyQ aggregates in touch neurons (Fig. 3c–f), we also examined exophers in a strain that co-expressed mSc::IFD-1 and mNG::HttQ74 (Fig. 5d). In these studies, we scored

exophers after 6 hr of food withdrawal, a culture condition that increases exopher production[20]. Of 30 total exopher events, 14 exophers included detectable mSc::IFD-1, establishing that tagged intermediate filament mSc::IFD-1 can be extruded in exophers in nutrient-stressed mNG::HttQ74-expressing neurons (Fig. 5e) (image of polyQ and mSc::IFD-1 extrusion in Fig. 5d). Our observation that mSc::IFD-1 and GFP::IFD-1 are included in some, but not all, of the exophers formed in the two different backgrounds (~15% mCherry; ~ 50% HttQ74), make the important point that IFD-1 extrusion is not a mandatory requirement in all exophergenesis events.

## Ejected IFD foci colocalize with mNG::HttQ74 in most extrusion events

To assess the IFD-1 status relative to ejected cargo, we measured the frequency of mNG::HttQ74 overlap with the mSc::IFD-1 that were extruded in exophers (*i.e.*, in exophers that included mSc::IFD-1 puncta/ae). Of the 14 mSc::IFD-1-positive exophers we reported in Fig. 5e, 11 exophers contained species in which mSc::IFD-1 and mNG::HttQ74 colocalized to the same puncta; 3 exopher events had mSc::IFD-1 foci in exophers(s) that lacked co-enrichment of mNG::HttQ74 (Fig. 5e). We conclude that IFD-1-aggresome-like organelles hosting HttQ74-cargo are often components of extruded exophers.

In a complementary analysis alternatively focused on cases in which HttQ74 aggregates leave the cell, we looked at the same 30 exopher events with an initial scoring of the mNG::HttQ74 reporter, and later checked for co-association of the IFD reporter. We found that 22 of the 30 mNG::HttQ74-identified exophers contained at least one identifiable mNG::HttQ74-enriched punctum (Fig. 5f). Of those 22 exopher events that included mNG::HttQ74 punctae, we asked whether the ejected HttQ74 was accompanied by IFD-1. We found 11 out of 22 exophers also displayed mSc::IFD-1 colocalized with mNG::HttQ74 (Fig. 5f). Data indicate that HttQ74-aggregates are often extruded into exophers in IFD-1+ aggresome-like organelles, although HttQ74 can leave the soma in the exopher compartment nearly as frequently without co-IFD-1 association. We conclude that IFD-1 association is not strictly required for HttQ74 extrusion.

## hNFL can partially complement *ifd-2(Δ)* in exopher formation

Mammalian intermediate filament proteins are integrated into the neuronal proteostasis network and can associate with juxtanuclear aggresomes[3]. Roles for mammalian IFs in neuronal aggregate extrusion have not been reported, but our *C. elegans* observations raise the possibility that mammalian neuronal IFs might act in exopher-related biology[13,16]. To begin to address this possibility, we asked if a human neuronally-expressed intermediate filament protein can complement an *ifd(Δ)* for exopher production deficits.

We focused on testing human neurofilament light chain (hNFL) because human NFL is clinically evaluated as a circulating blood/CSF biomarker of neuronal injury and neurodegenerative disease[36], despite mysteries surrounding the mechanistic origins of NFL that circulates. hNFL is a neuron-specific intermediate filament protein and exhibits ~24/45% sequence identity/similarity to *C. elegans ifd-1* and *ifd-2*.

We optimized the human NFL gene for expression in *C. elegans*[46] and examined the subcellular distribution of the tagged mNG::hNFL expressed from a single copy insertion specifically in touch neurons. (Supplementary Fig. 15). mNG::hNFL concentrated to juxtanuclear foci that colocalize with *C. elegans* intermediate filament reporter mSc::IFD-1 (Fig. 6a). mScarlet::hNFL colocalizes with ubiquitin, mNG::UBQ-2 (Fig. 6b), and with mNG::HttQ74 (Fig. 6c) in aggresome-like compartments. Although the predominant localization of mNG::hNFL is to one or two punctate juxtanuclear locations (Supplementary Fig. 15) with both single copy and high copy mNG::hNFL reporters, we did note that FP-tagged hNFL could also be

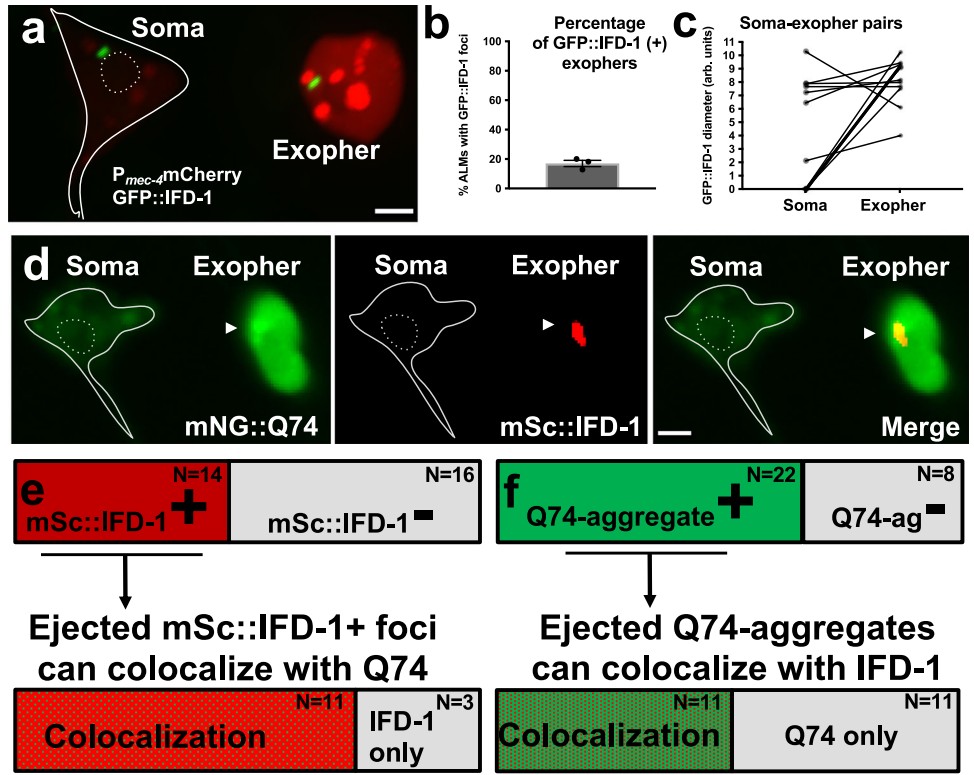

**Fig. 5 | IFD proteins can be extruded in exophers. a** IFD-1-positive puncta extruded in an exopher in a strain that expresses *bzIs166*[P*mec-4*mCherry] for touch neuron/exopher visualization. Shown is an adult touch neuron from a strain expressing *bzIs166*[P*mec-4*mCherry]; *bzSi3*[P*mec-7*GFP::IFD-1]. Soma outline is in white, nucleus is outlined with a white-dashed line. One GFP::IFD-1 focus remains in the soma, and one is ejected in the exopher. Scale bar = 2 um, representative of N > 10 GFP::IFD-1+ exophers. **b** We examined a strain expressing *bzIs166*[P*mec-4*mCherry]; *bzSi3*[P*mec-7*GFP::IFD-1] on Ad3 and determined how often GFP::IFD-1 was extruded in exophers. In total, we observed 61 ALM Ad3 exopher events, with 11 having GFP::IFD-1 included in the exopher, 3 trials, error bar is SEM. **c** We looked at neurons that ejected a GFP:IFD-1. Of 9 neurons in which GFP::IFD-1 was expelled in the exopher, 4 neurons ejected the only – and entire - GFP::IFD-1 organelle visibly present. Graphed as GFP::IFD-1 diameter in arbitrary units. **d** Shown is an adult touch neuron from a strain expressing *bzSi39*[P*mec-7* HttQ74::mNG]; *bzSi34*[P*mec-7*mScarlet::IFD-1] on Ad1 after a 6 hr fast. Soma outline is in white, nucleus is outlined with a white dashed line; arrowhead points to the concentrated HttQ74

punctum that colocalizes to the IFD-1 domain. Scale bar = 2 μm, representative of N > 10 exophers. **e** We examined a strain expressing *bzSi39*[P*mec-7*HttQ74::mNG]; *bzSi34*[P*mec-7*mScarlet::IFD-1] on Ad1 after a 6 hr fast to increase exopher yields, and asked how often mScarlet::IFD-1 was extruded in exophers. We observed 30 ALM exopher events, with 14 having mSc::IFD-1 included in the exopher. Next we asked how often the 14 mScarlet::IFD-1-positive exopher events also included at least one concentrated HttQ74 puncta overlapping with mSc::IFD-1. In 11/14 ALM IFD-1-positive exophers, there was at least one mNG::HttQ74 punctum colocalized with mSc::IFD-1. **f** We examined the same exopher events as described in (**e**) above. We determined how often HttQ74 aggregates were extruded in exophers. Out of *N* = 30 ALM exopher events, 22 exopher events had at least one HttQ74 aggregate included in the exopher. We asked how often mNG::HttQ74 positive exopher events (*N* = 22) included species where HttQ74 is a cargo of the mSc-aggresome-like organelle to find 50% (11/22) of ALM HttQ74-positive exophers had at least one incidence of HttQ74 including mSc::IFD-1.

distributed more broadly in the neuron as compared to *C. elegans* FP::IFDs, also appearing in the axon, nucleus, and in filamentous patterns in the cytoplasm in some neurons (Supplementary Fig. 15). Nonetheless, *C. elegans* IFDs and hNFL clearly localize similarly to aggresome-like compartments (Fig. 6a). Akin to what we documented for the *C. elegans* IFD-aggresome-like organelle (Fig. 5), we found that hNFL inclusions are expelled within approximately half of *C. elegans* exophers (Fig. 6d).

We next wanted to investigate neuron-autonomous function of hNFL in exophergenesis. We introduced the P*mec-7*mNG::hNFL single copy transgene into an *ifd-2(Δ)*; mCherry background and compared exopher production in *ifd-2(Δ)* vs. *ifd-2(Δ)*; P*mec-7*mNG::hNFL. Single copy P*mec-7*mNG::hNFL can partially rescue *ifd-2(Δ)* exopher deficits (*P* = 0.0046; Fig. 6e). The mNG::hNFL single copy construct, similarly to tagged IFD constructs, does not significantly elevate or suppress exopher levels on its own, indicating that the rescuing activity is unlikely to be caused by hNFL aggregation (Supplementary Fig. 16). hNFL rescue of *ifd-2(Δ)* exopher deficits supports the proposition that human neuronal intermediate filament proteins play a conserved role in exopher biology.

In sum, our finding that human hNFL shows overlapping subcellular localization with *C. elegans* IFD and aggresome-like organelle components (Fig. 6a–c), is expelled from the soma in exophers (Fig. 6d), and can partially compensate for *ifd-2* expulsion function in vivo (Fig. 6e) raise the possibility that mammalian IFs may play roles in cellular extrusion that are conserved from nematodes to humans, inviting the re-examination of mammalian aggresome biology in the context of mechanisms of human aggregate spread.

## Discussion

*C. elegans* intermediate filament proteins IFD-1 and IFD-2 concentrate into stress-responsive juxtanuclear foci that are analogous in position and composition to that reported for mammalian aggresomes. IFD-1 and IFD-2 also act in proteostressed neurons to mediate efficient extrusion of cytoplasmic contents in large exopher vesicles (Summary in Supplementary Fig. 17). Conservation of the associated biology is supported by our finding that human neuronal intermediate filament protein hNFL, expressed in *C. elegans* neurons, localizes to the IF-aggresome-like organelle, can partially substitute for *C. elegans ifd-2* in exopher production, and can be extruded in

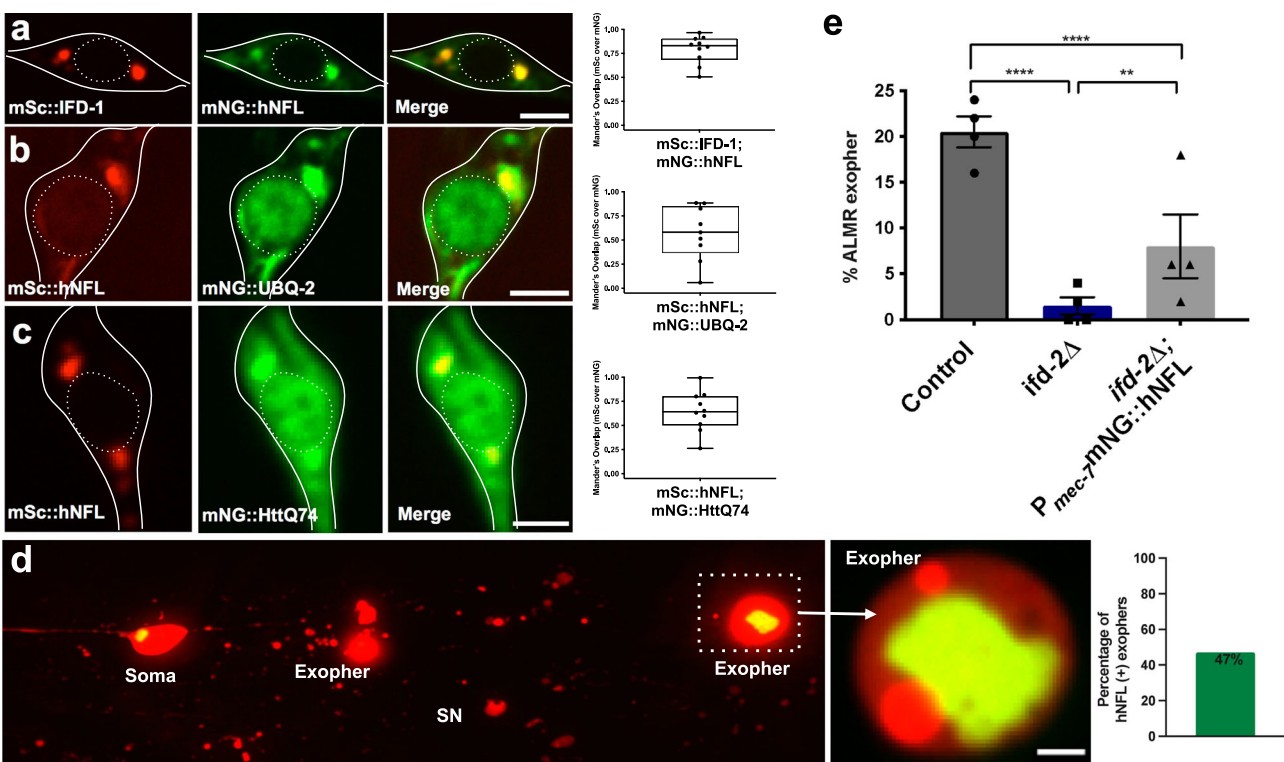

**Fig. 6 | IF roles in aggregate collection and neuronal aggregate expulsion may be partially conserved. a–c** We optimized the coding sequence of human Neurofilament Light Chain, hNFL (NM_006158.5), for *C. elegans* expression codon biases (see "Methods"), tagged with mNeonGreen fluorophore and expressed under the *mec-7* touch neuron promoter. Scale bar = 2 μm. **a** Strain is expressing *bzEx269*[P*mec-7*mNG::hNFL] and *bzSi34*[P*mec-7*mSc::IFD-1]. Image is representative of N > 20 ALMs. Graphed as Mander's overlap coefficient of mScarlet channel over mNeonGreen channel, *N* = 10 ALMs graphed. Boxes indicate coefficient range; minimum (0.51), maximum (0.97), and mean (0.80) coefficient. **b** Adult touch neuron expressing *bzEx313*[P*mec-7*mNG::UBQ-2]; *bzSi43*[P*mec-7*mSc::hNFL]. Image is representative of N > 10 ALMs. Mander's overlap coefficient graphed with boxes indicating coefficient range minimum (0.06), maximum (0.84), and mean (0.57). **c** Representative adult touch neuron expressing single copy *bzSi39*[P*mec-7* mNG::HttQ74]; *bzSi43*[P*mec-7* mSc::hNFL]. Image is representative of touch neurons that display visible mNG aggregation, N > 10 ALMs. Mander's overlap coefficient plotted; *N* = 10 ALMs that display visible mNG aggregation. Graphed as Mander's overlap coefficient; boxes indicate coefficient range; minimum (0.23), maximum

(0.99), and mean (0.73). **d** hNFL-labeled aggresome-like foci can be extruded from touch neurons in exophers; representative image (N > 10 ALM exophers). Strain expresses *bzEx311*[P*mec-7*mNG::hNFL] with *bzIs166*[P*mec-4*mCherry] in the *ifd-2(bz187)* background. Image shows ALMR neuron with three exophers. Two exophers are positioned in the middle of the image surrounded by scattered mCherry debris, starry night phenotype (SN) from hypodermal exopher digestion[6]. The third exopher pictured, a mature distant exopher, contains a large mNG::hNFL inclusion, shown in focus and zoomed in. The exposure is altered to show details. Out of N = 34 Ad2 ALM exophers from a strain expressing *bzSi48*[P*mec-7* mNG::hNFL]; *bzIs166*[P*mec-4*mCherry], 16 exophers included mNG::hNFL puncta. **e** Touch neuron expressed single copy P*mec-7*mNG::hNFL partially rescues the exopher suppression phenotype of the *ifd-2(bz187)* mutant. We measured Ad2 exophers in ALMR. *bzIs166*[P*mec-4*mCherry] compared to *bzIs166*[P*mec-4* mCherry]; *ifd-2(bz187)* (****$P = 3.9 \times 10^{-9}$) and *bzIs166*[P*mec-4*mCherry]; *ifd-2(bz187)* compared to *bzIs166*[P*mec-4*mCherry]; *ifd-2(bz187); bzSi48*[P*mec-7* mNeonGreen::hNFL] (**$P = 0.0046$), *N* = 200 ALMR per strain, CMH test, 4 trials.

exophers similarly to what we find for *C. elegans* IFDs. Our data hold implications for considerations of proteostasis, vesicle extrusion, and neuronal health.

The original description of mammalian aggresomes (focused in cultured CHO and HEK cells) defined multiple aggresome features, including prominent juxtanuclear concentration of aggregates with distinct localization from Golgi and lysosomes, association with intermediate filaments under proteostress conditions (aggregate-prone protein over-expression or inhibition of the proteasome), colocalization with ubiquitinated and heterogeneous cargo, and requirements for microtubule and dynein motor integrity for biogenesis[3]. Follow up studies established that p62/SQSTM1(SQST-1) and components of the 14-3-3(FTT-2)/BAG3/Hsc70(HSP-1) complex associate with small foci that move into the large perinuclear aggresome-like organelle[26,47]. We find that most of these mammalian aggresome features are shared with the IFD-associated puncta in proteo-stressed *C. elegans* touch neurons, including collection of aggregated polyQ protein, ubiquitin enrichment, concentration of p62/SQSTM1(SQST-1) and 14-3-3(FTT-2)/Hsc70(HSP-1), and a requirement for intact microtubules and dynein activity to collect materials

into one or two juxtanuclear sites. Although aggresome-like structures associated with the expression of human neurodegenerative disease proteins[48] or ROS stresses[19] have been reported for *C. elegans* cells, our multi-component characterization of *C. elegans* neuronal aggresome-like organelles definitively establish that aggresome biogenesis mechanisms are at least partially conserved from nematodes to humans. *C. elegans* proteostressed neurons thus can serve as a validated model for deciphering the relative role of aggresomes in maintaining neuronal proteostasis and health in an in vivo context.

Limited ultrastructural data inform on mammalian aggresomes[3,5,24,25,49–52]. Our EM data for proteostressed *C. elegans* neurons suggest an initial early association of short IFs within the concentrated central granular material positioned near the nucleus, followed by maturation featuring a loose discontinuous IF meshwork that surrounds proteinaceous content, and eventually features longer more continuous fibers at the periphery (Supplementary Fig. 2). Assembly of a loose cage suggests feasibility of the proposal that once the IF mesh assembles with initial aggregates to nucleate the aggresome compartment, cargo might be later able to diffuse directly into and around the aggresome[53].

At the cell biological level, a striking observation is that as polyQ accumulates in the neuron over time, the IF/polyQ association progressively appears imbalanced, with the polyQ domain appering to outgrow the IF domain, which no longer surrounds the polyQ domain (Supplementary Fig. 4). Thus, under conditions of high expression of neurotoxic proteins, IFDs may be limited in their capacity to fully encase cargo like aggregated polyQ128, raising the possibility that imbalances in IF or aggregate levels might alter function/capacity of the IF-associated aggresome-related organelle. The cellular consequences of IF/aggregate proportion changes are not clear, and the question as to whether these structures correspond to an aggresome uncoating step[54,55] prior to content degradation via autophagy remains to be addressed. Changing patterns of polyQ concentration have been reported in other in vivo studies in *C. elegans*[56] and HEK cells[53], suggesting dynamics may be more common features of aggresome biology than previously thought.

Our data show that aggresome-associated proteins that collect and concentrate with protein aggregates can modulate exophergenesis levels. In particular, the documentation of multiple IFs and aggregate collection factors (IFD-1/IF, IFD-2/IF, FTT-2/14-3-3, HSP-1/Hsc70) localized to the same soma compartment with shared functional capacity to modulate exopher extrusion levels support an unanticipated intersection between aggresome-like organelle formation and exopher extrusion.

Our genetic observations address multiple hypotheses regarding the nature of the aggresome/exopher connection. First, IFDs exert neuron-intrinsic influences on exophergenesis. Although *ifd-1* or *ifd-2* deficiency might be hypothesized to induce or inactivate a systemic stress response, our data support a touch neuron-autonomous role in extrusion efficiency for IFD-1 and IFD-2, rather than general roles in animal or neuronal health. Second, aggresome dimension, per se, does not suffice to trigger expulsion. Measures of the aggresome-like compartment relative to extrusion do not indicate a strong correlation of aggresome-like compartment dimensions with exopher levels (Fig. 5c, Supplementary Fig. 13), supporting that the exophergenesis trigger is not exclusively activated by aggresome compartment size. Third, modulation of *ifd* expression levels is unlikely to drive exopher production. IFD over-expression is not sufficient to elevate exophergenesis and IFD proteins are necessary but not sufficient to drive exopher production (Supplementary Fig.10, Supplementary Fig. 11). Finally, IFs are not required partners for extruded aggregates, nor are IF-aggresome-like organelles essential components of exophers (Fig. 5). Defining molecular details of the IF/exopher relationship remain an important area for future study.

Exopher-like mechanisms involving IFs might influence a range of disease processes and underlie the release of neurodegeneration biomarkers. Heritable mutations in neurofilaments can cause several neurological diseases including Giant Axonal Neuropathy, Charcot-Marie-Tooth (CMT) disease, Amyotrophic Lateral Sclerosis (ALS), and Parkinson's disease (PD)[57]. Neurofilaments have also been widely implicated in neurodegenerative disease pathology due to their concentrated presence within characteristic inclusions, such as Parkinson's disease Lewey bodies[38,58]. Our findings raise the possibility that IF-modulated content expulsion under neuronal stress conditions might be an unexpected contributor to disease pathology in these disorders.

From another perspective, it is interesting that the presence of human neurofilament light protein, NFL, in CSF and blood is clinically measured as a biomarker for neuronal disease and injury[58]. We found that IFD-1 and IFD-2, sharing ~ 24% identity and ~ 43% similarity with human NFL, can be expelled within exophers in a significant fraction of exopher events. Thus, *C. elegans* exopher studies reveal a previously unappreciated path through which IFs are released from proteo-stressed neurons and raise the possibility that the commonly used NFL

biomarker of human neuronal injury and disease might also reflect active neuronal exopher expulsion mechanisms. Potentially supporting this idea, intermediate filaments are also a major reported protein component of exopher-like extruded material from stressed cardiomyocytes[13,14]. Research into IFD or NFL expulsion in *C. elegans* may therefore both help explain basic mechanisms promoting aggresome- and Lewy body-like inclusion pathology that results in disease, and illuminate understanding of clinical biomarkers used to assess pathology.

The identification of specific *C. elegans* intermediate filaments that influence exopher production links to the increasing documentation of IF roles in proteostasis. Interestingly, studies on the mouse vimentin knockout reveal that IF protein vimentin is not necessary for either basic cellular function or aggregate collection at the juxtanuclear aggresome[59], similar to our *ifd* observations in the *C. elegans* touch neuron model (Supplementary Fig. 18). As in *C. elegans*, redundancy among intermediate filaments might mitigate effects. In mammals, vimentin has been implicated in stress responses that include heat shock, wound healing, and Alzheimer's disease challenges[60]. Adult mammalian neural stem cells rely on vimentin for activation of proteostasis and aggregate elimination as the NSCs exit from quiescence[59]. In the NSC model, vimentin is implicated in spatial organization of proteasomes and autophagosomes to the aggresome domain, possibly anchoring machinery that promotes degradation of aggresome contents[59]. Detailed study of these processes in multiple models may further underscore commonalities and clarify mechanism.

That aggresome-like organelles can be cleared from neurons by extrusion holds implications for mammalian proteostasis. IFD-marked aggresome-like foci can frequently leave the neuron together with cargo (Fig. 5). In about half the cases in which IFD left in an exopher, the entire aggresome-like organelle content of the soma exits in the exopher (Fig. 5c). The point we underscore is that whole aggresome-like organelles can be removed from the neuron of origin for remote degradation, a new observation in the field.

Exophers may represent a conserved mechanism used to clear aggresomes. The fundamental mechanisms by which proteo-stressed neurons manage toxic aggregates and damaged organelles remain a central question in neurodegenerative disease[12]. In mammals, processes strikingly similar to *C. elegans* exophergenesis have recently been described for HeLa cells[61], PC12 cells[61], and in vivo extrusions from mouse cardiomyocytes[14]. Moreover, in vivo transcellular extrusion of mitochondria from retinal ganglion cells for degradation by neighboring cells is reminiscent of exopher-like biology, and transcellular extrusion and degradation is increased in AD-model mouse brain[62] and has been reported in human neurodegenerative disease brain. Emerging data thus highlight a newly identified cellular strategy for offloading aggregates and organelles for remote management that may operate across phyla. Although molecular links among the extrusion events across species remain to be rigorously established, initial observations are consistent with the hypothesis that exopher-like responses are within the capacity of a variety of stressed cells in a range of tissues across organisms (see review on heterophagy, ref. [15]).

By extension, our findings that *C. elegans* exophers can clear aggresome-like organelles suggest a new perspective on the cell biology of aggresomes and raise the possibility that aggresome-related mechanisms influence human neuronal aggregate spreading. Along these lines it is interesting that elevated expression of mammalian 14-3-3θ can promote alpha synuclein extrusion[63], akin to our finding that 14-3-3 protein FTT-2 is required for efficient exophergenesis and can drive enhanced exopher production when over-expressed in *C. elegans*. Overall, our data invite a fresh look at how mammalian aggresomes are cleared and how exopher-like pathways contribute to spreading mechanisms.

## Methods

### Strains and media

We cultured *C. elegans* strains at 20 °C with standard methods[7]. We fed animals *Escherichia coli* OP50 unless we initiated RNAi or a 6-hr fasting protocol to induce proteostress-exophers. Extrachromosomal arrays and drug-resistant strains were harbored on antibiotic selection plates (Hygromycin B or G418) to ensure homozygosity. For exopher measurements, L4 animals were removed from selection plates and placed onto standard OP50 plates for later observation.

### Strains

New transgenic alleles are provided in figure legends and in the strain list in Supplementary Data 1. Reporters for FP::IFD were expressed from the touch neuron specific *mec-7* promoter to focus on cell autonomous effects; single copy reporters from the native *ifd-1* and *ifd-2* promoters did not provide adequate neuronal signal in touch neurons for visualization. P$_{mec-7}$FP::IFD constructs were effective at complementation of the cognate *ifd* alleles and were neither dominant-negative nor exopher-inducing on their own. Animals expressing *igIs1*[P$_{mec-7}$YFP P$_{mec-3}$HttQ128::CFP] must be manually selected for CFP expression to avoid silencing of the CFP, which we have not been able to easily reverse (high copy number silencing has been documented[64]). PolyQ constructs are quite challenging to generate due to the extent of repeat sequences therein; still in an attempt to make and study a more co-operative polyQ in touch neurons, we constructed a single copy *bzSi39*[P$_{mec-7}$mNG::HttQ74] allele. The polyQ protein aggregates minimally.

See Supplementary Data 1 for strain list.

### Plasmid constructions

New allele information is described in Supplementary Data 1. We created single-copy integration using MiniMOS technology and pCFJ1662 hygromycin or pCFJ910 G418-resistance conferring backbones[65]. For extra-chromosomal array transgenesis and selection we used hygromycin or G418 selection by exposing animals to selection plates. To make selection plates, we used 5 mg/mL for hygromycin or 32 mg/mL for G418 and added 200 µl per 6 cm OP50-seeded plate.

See Supplementary Data 1 for RNAi bacteria clone sequences.

### Human homolog optimization and expression

Both IFD-1 and IFD-2 are most closely related to the named mammalian (taxid: 9606) neurofilament light polypeptide (NP 006149.2) with e-values of 2e$^{-15}$ and 3e$^{-13}$ for IFD-1 and IFD-2 respectively, sharing 24% identity. To optimize human genes for *C. elegans* expression based on codon and intron bias and *C. elegans* amino acid to tRNA ratio, we used *C. elegans* Codon Adapter GGA Software[46]. Nucleotide coding regions of mammalian proteins are provided. NCBI Reference Sequence: NM 006158.5 for *homo sapiens* neurofilament light. Optimized hNFL sequence available in Supplementary Data 1.

### CRISPR-CAS9 genomic *ifd-2* deletion

*ifd-2(bz187)* is a deletion of base pairs 816271-815275. The first three exons are affected by the genomic deletion which eliminates 996 base pairs, including the *ifd-2* initiation codon. Guide RNA and repair oligo available in Supplementary Data 1.

### RNAi

We achieved gene inactivation by feeding animals RNAi bacteria (HTT115) from L4-Ad2 or for 2 generations as noted, using standard methods. We performed RNAi experiments in strains in which neurons are RNAi-enhanced via SID-1 expression[6,7] required for RNAi effects in neurons[45]. We used two transgenes for neuronal RNAi enhancement, initially working with the P$_{mec-18}$sid-1 transgene, which enhances touch neuron RNAi and later pan-neuronal P$_{rgef-1}$sid-1, which consistently supported strong knockdown (gift from the lab of Dr. Malene Hansen).

Some strains expressed pan-neuronal *sid-1* in the background of a *sid-1* null mutant, such that only transcripts in neurons could be targeted for degradation. Whenever possible, we confirmed RNAi knockdown effects using alternative approaches.

### Age synchronization

To synchronize animals, we selected L4 stage hermaphrodites and transferred them to test plates. The day after moving was considered adult day 1, Ad1. We scored animals on adult day 2 (Ad2) unless otherwise noted. For scoring exophers, we measured animals on plates using a Kramer pseudo-stereo dissecting scope (20x objective) and immobilized using 100ul of 10 mM tetramisole; exophers were also visible in live animals without anesthetic.

### Exopher features for scoring

Touch neuron exophers are readily visible at 400X total magnification with high power dissecting microscopes. We performed exopher characterization methods as described in detail[7]. We note that baseline exopher level in the P$_{mec-4}$mCherry strain is generally in the range of 5-15% of animals with ALMR exophers on Ad2. The general or overall (mean) variability of other genetic backgrounds can differ. It is critical to *always* compare control and experiment on the same day because of the baseline variability—a test outcome must be compared to the background of its paired control.

### Microscopy

Most high resolution fluorescence micrograph data were captured on Zeiss Axiovert Z1 microscope equipped with X-Light V2 Spinning Disk Confocal Unit (CrestOptics), 7-line LDI Laser Launch (89 North), Photometrics Prime 95B Scientific CMOS camera, using Metamorph 7.7 software. A ×100 oil immersion objective, ×63 oil immersion, or ×40 objective was used.

### Fluorescent colocalization analysis

We imaged touch neurons for colocalization experiments. Where needed, we used DIC confirmation to identify touch neurons of interest. For mNG::HttQ74 and HttQ128::CFP colocalization, we used only neurons in which visible mNG/CFP-aggregation was observable in the cell (as opposed to dim cytosolic mNG signal or absent CFP signal) for colocalization calculations. We analyzed thresholded multi-color image channels using ImageJ FIJI with JACoP and Coloc2 analysis software to calculate Pearson's and Mander's overlap coefficients.

We used Pearson's coefficient to quantitate colocalization of organelle tags and calculated Mander's overlap of thresholded double-channel images by using the channel relevant to tagged IFD over the channel of the other protein tag. We manually analyzed images in each 0.2 µm Z-stack for colocalization often observing Z-stacks laterally in 3D projection for 3-dimensional observation.

We did note that IFD inclusions were often found next to Golgi or lysosomes, and thus the possibility that the IFD compartments might interact with these organelles, potentially via transient associations, should be entertained in future study.

### Soma GFP::IFD-1 focus measurement

We imaged whole touch neurons using 0.2 µm Z-stacks. We measured the largest diameter of each inclusion by calculating the longest pixel length using ImageJ software. We omitted neurons with the presence of an exopher(s) from GFP::IFD-1 size calculations. We calculated the number of IFD inclusions using a standard threshold.

### Drug assays

For proteostasis drug treatments, we dissolved MG132 (Sigma-Aldrich C2211) and Spautin-1 (Sigma-Aldrich SML0440) in DMSO at 10 mM concentration and administered by placing 40 µL of each solution over the bacterial food lawn. We exposed animals to drug-plates from

L4-Ad2 or Ad3. For DHC-1::degron AID-system treatments, we dissolved auxin in ethanol to 0.4 M. We added 25 µL 0.4 M dissolved auxin and 175 µl distilled water. We pipetted 200 µl solution, per plate, on top of well-seeded OP50 plates. We exposed both control and experimental animals to auxin plates from L4-Ad2 - only the experimental animals harbored $bzSi6$[P$_{mec-7}$TIR1::TagBFP] necessary for auxin-inducible degradation[30].

## Osmotic stress

Osmotic stress is associated with proteostasis disruption[66]. For osmotic stress assays we used NGM agar with 450 mM NaCl. We stored plates overnight at room temperature and subsequently inoculated with concentrated OP50 overnight culture; after overnight incubation at room temperature plates were ready for assay. We started osmotic stress experiments by transferring L4 larvae onto the bacterial lawn followed by an overnight incubation. Subsequently, we washed in recovery buffer (M9 buffer with 150 mM NaCl) and transferred to normal NGM plates. The viability of each animal after overnight exposure to NaCl was scored using mechanical stimulation. We graphed the percent survival of the population after 24 hours.

## Oxidative stress

Oxidative stress is associated with proteostasis disruption[67]. For oxidative stress we used 200 mM paraquat in NGM agar. We stored plates overnight at room temperature and subsequently inoculated with concentrated OP50 overnight culture. Following overnight incubation at room temperature plates were ready to use. We started oxidative stress experiments by transferring L4 larvae onto the bacterial lawn. We scored animals hourly for viability using mechanical stimulation with a platinum wire. Control plates did not contain paraquat. We graphed a survival curve as the percent surviving each time point until all animals were dead.

## Development

We measured post-embryonic development time (hours from egg to young adult stage) by transferring eggs to an NGM plate. To do so, we performed a 50-gravid animal egg lay lasting one hour upon a new NGM plate. In total, 60 hr after the egg lay, we checked plates every hour and the percentage of progeny to develop was assessed; we removed young adults after counting. We calculated time of development as the average number of days animals needed to reach adulthood. We graphed the percentage of the population that had reached the adult stage at each timepoint. For significance tests we used a standard two-tailed $t$ test comparing the mean value of three trials.

## Fertility

We measured brood size by placing individual L4 stage worms onto NGM plates. We transferred animals to new plates daily for three days, allowing progeny to develop to adulthood before paralysis and quantification.

## Lifespan

We measured lifespan on NGM plates, transferring animals to fresh plates each day until day 5 of adulthood (Ad5) and after every four days thereafter. We assessed viability every two days as response to gentle prodding with a platinum pick; movement response was scored as alive. Animals that either had internal hatching of progeny or expulsion of internal organs were not counted as deaths. We excluded animals that could not be located on the plate from statistical analysis.

## Mechanosensory touch function

We lightly stroked animals with a fine-hair pick behind the pharynx five times and scored a positive response as an animal movement in the reverse direction. For example, if an animal responded 2 out of 5 times, the animals has a 40% response rate. We graphed the percentage response rate of each animal.

## GST-4::GFP reporter assay

GST-4 is a glutathione S-transferase that facilitates the Phase II detoxification process. $gst-4$ expression is regulated by $daf-16$ and $skn-1$ and is a common reporter of cytosolic stress. We mounted animals expressing $dvIs19$[(pAF15)P$_{gst-4}$GFP::NLS] +/- $ifd(\Delta)$ on microscope slides for 40x imaging and reported maximum intensity per animal in arbitrary units.

## Electron microscopy

We prepared mCherry animals for TEM analysis by high pressure freezing and freeze substitution (HPF/FS), and followed a standard to preserve ultrastructure. After HPF in a Baltec HPM-010, we exposed animals to 1% osmium tetroxide, 0.1% uranyl acetate in acetone with 2% water added, held at −90 °C for 4 days before slowly warming back to −60 °C, −30 °C, and 0 °C, over a 2 day period. We rinsed the samples several times in cold acetone and embedded the samples into a plastic resin before curing them at high temperatures for 1–2 days. We collected serial thin sections on plastic-coated slot grids and post-stained them with 2% uranyl acetate and then with 1:10 Reynold's lead citrate, and examined with a JEOL JEM-1400 Plus electron microscope. By observing transverse sections for landmarks such as the 2nd bulb of the pharynx, it was possible to reach the vicinity of the ALM soma before collecting about 1500 serial thin transverse sections.

We identified 29 ALM somata in serial sections from 23 blocks. We selected 12 candidate cells that were initially identified using light microscopy to contain touch neurons in the bud phase of exophergenesis. We imaged juxtanuclear regions of interest at high-magnification in serial sections, and we took high power electron tomograms (15 K or 20 K) in those neighborhoods, comparing regions showing either aggresome-like structures, mitochondria, or large lysosomes. We closely examined those regions for evidence of intermediate filaments surrounding and/or lying inside of the organelles. We used IMOD and TrakEM software for 3D analysis of the regions of interest, and for model-making.

## Statistical analysis

Each trial is graphed as a percentage of ALMR exopher occurrence (binary); we used Cochran−Mantel−Haenszel (CMH) analysis, which considers the number of exophers (binary) and the number of observations (ALMRs) in a paired manner with the control to generate an estimate of association. $P$ values were calculated using at least three or more biological trials. For IFD measurements, we used two-tailed $t$ tests. We analyzed the proteostress assays and comparison of number of IFD-inclusions by $t$ tests or one-way ANOVA (with Dunnet's post-test) as noted.

## Blinding

For exopher measurements, we recorded strain information in a non-visible location. The data were unblinded following completion of the experiment. For blinding microscopy analysis, we mounted and imaged all animals on a slide for a given strain. Post-acquisition, the file names were assigned a random number and re-coded. The images were unblinded following analysis of the entire data set.

## Reporting summary

Further information on research design is available in the Nature Portfolio Reporting Summary linked to this article.

# Data availability

IFD sequences with the named mammalian (taxid: 9606) neurofilament light polypeptide (NP 006149.2). We used NCBI Reference Sequence, NM 006158.5, for *homo sapiens* neurofilament light, to design constructs for *C. elegans* expression optimization. Sequences are available in Supplementary Data 1. Source data are provided with this paper.

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

## Acknowledgements

We thank past and present members of the Exopher Club for exopher-related discussions, and members of M.D. and B.D.G. laboratories at Rutgers University. We thank H. Ushakov for injections that created some transgenic lines. We thank Yunpeng Xu for sharing detailed control studies on neuronal RNAi efficacy. We thank Florian Geissler/Rudolf E Leube for providing plasmid DNA templates for IFD promoter amplification[68] and Malene Hansen, Xing She, and Yongzhi Yang for neuronal RNAi strains with enhanced efficacy and RNAi-related discussions. Some strains were provided by the CGC (which is funded by NIH Office of Research Infrastructure Programs (P40 OD010440) and the OK *C. elegans* Deletion Mutant Consortium). We used BioRender to create some cartoon figures. Research was supported by the National Institutes of Health under award numbers R37AG56510, R01AG047101, 5R01GM135326, and 1F31AG066405. K.N. and D.H.H. were supported by NIH OD010943. Content is solely the responsibility of all authors and does not necessarily represent the official views of the National Institutes of Health.

## Author contributions

M.L.A., J.F.C., R.A., S.A., I.M., A.J.S., and R.J.G., conducted and designed experiments, along with M.D., M.L.A., and B.D.G, who wrote the manuscript with input from J.F.C., R.A., and R.J.G. G.B. provided some plasmid constructions. K.C.N. and D.H.H. carried out electron microscopy fixation, interpretation, and analysis.

## Competing interests

The authors declare no competing interests.
