## [Peer Review File · Nature Communications]

Reviewer #1 (Remarks to the Author):

The revisions are fine and address our main comments. The smaller points listed below could be considered by the authors for the final version.

Some extra explanation:

-In the cell autonomous section of the paper (from line 242), it is shown that a single-copy GFP-tagged IFD-2 transgene, under its native promoter, can rescue the *ifd-2* ko phenotype on exopher formation. It is not clear why this was not also tested for *ifd-1*. The authors could comment on this in the manuscript.

-The intestinal expression of fluorescent tagged-IFD-2 shows a clear trend and an almost significant partial rescue in the exopher deficiency of the *ifd-2* ko worms (Supplementary figure 10). These results could very well be significant with increasing the n. Even when *ifd-2* is involved in exophogenesis acting from neurons, a small contribution from IFD-2 in exopher formation from the gut can't be rule with the current data. This should be briefly discussed in the text.

Small edits:

-The x axis legends in Figure 1D and 1E for the rescue should denote "Pmec-7GFP::IFD-1" and "Pmec-mNeonGreen::IFD-2" to make more clear that these experiments were done with exclusive expression in touch neurons. Alternatively, this could be added in the figure's titles.

-At the end of line 240, there was a statement in the previous version to acknowledge that other proteins than IFD-1 and 2 are likely involved in exopher production, giving that the double mutant is still able to produce them. It would be good to keep this statement: "Because some exopher production is evident when *ifd-1* and *ifd-2* are both absent, we infer that a redundant activity or a parallel pathway must also contribute to exopher formation."

- In Supplementary Fig 8A, the legend says the merged comes from GFP, mCherry and DIC channels. The DIC channel is missing. Also, giving the multiple single and multicopy controls throughout the text, which makes the story very difficult to follow in certain points, it would be good to add "OE" in the title of figures 8A and 8B. Figure 8D shows endogenously tagged IFD-2 but the legends describes "*ifd-1* gene *ifd-1(bz477)*", please clarify this.

-The section starting from line 280, to test gene dosage effects does not correlate well with the subfigures shown in Supplementary Figure 6B. The text describes first Sup. Fig 6B and the effect of *ifd-2* OE into *ifd-1* ko animals, next Sup. Fig 6C and finally Sup. Fig 6A. Please describe the data in a more logical order that correspond better with the figures or adapt the figure. Also, only the cross complementation of IFD-2 OE in the *ifd-1* ko background is shown, but the vice versa mentioned in the text, which should be the IFD-1 OE in the *ifd-2* ko background is missing. Is the *ifd-1* ko; IFD-2 OE significantly different than the *ifd-1* mutant in figure 6A?

-The statement "Human neuronal intermediate filament protein hNFL can complement *C. elegans ifd-2* deletion to promote exopher formation" should be rephrased as "can partially complement..."

- Supplementary figure 22, although presented in the rebuttal letter to further support the neuronal effect of FTT-2, is not referred, or explained to in the text.

Reviewer #2 (Remarks to the Author):

This manuscript is a mish-mash of experiments showing a multitude of factors regulating aggresome formation in neurons of *C. elegans*, but the manuscript is unfocused and instead of describing how IFs regulate exopher extrusion (the title of the paper) it meanders over various parameters of aggresome

formation that while interesting should be removed from this paper and described in its own manuscript.

The figures are similarly a mixture of information, often showing non-relevant things while omitting critical points. The approach seems to be to drown the reviewer in data, irrespective of whether they are or are not relevant to the story. Many critical figures have been relegated to supplemental, presumably to overcome the space limitations but, really 24 supplemental figures! Moreover, many of those figures are actually essential to the point of this manuscript while some of the figures or figure panels that are in the main manuscript don't add to the main thrust of the story.

In its current form, the results and the writing are so confusing that in the end it's almost impossible to decipher what has been fully documented, i.e. which exopher cargoes have been completely analyzed to show how they form exophers, whether they are surrounded or just adjacent to IFs, whether IFs and other regulators influence their expulsion from cells, and how the relative distribution of cargo and IFs within the cellular aggregates changes/doesn't change in the expelled exophers. Some of the data are probably dispersed in different figures and supplemental figures, but this reviewer had a hard time figuring out what exactly has been done and it is likely that readers would be similarly confused.

Additional issues:

1. Figure 1A – add what we are looking at - is that mCherry? And what is mCherry and why does it aggregate?; B) should be removed and replaced with a series of IFs/EMs/movies showing the formation of an exopher; should show IFs with mCherry – do the IFs surround the aggregate or are interspersed with the cargo or something else? – this should be shown for all exopher cargo described in this manuscript
 2. Figure 2 – A-C and E-K are irrelevant and should be in another paper
 3. Figure 3 – completely irrelevant to this paper and should be moved into another paper on aggregates formation
 4. Figure 4 – A and B should be in Supplemental; must show exophers with this cargo in a figure; should also show the Htt cargo in this figure and document and quantitate exophers (with or without IFs) containing Htt in this figure;
 5. Figure 5 – B is irrelevant and fits into another paper on aggregates formation; D – label what is red and what is green; why are these IFs not surrounding the cargo in the exopher– when did the remodeling occur since they were together in the cell in Figure 5C?
- Include in the paper in the appropriate place the data from supplemental figures 5, 14, 17, 18, 19, 21, and 22 – those are actually the interesting and informative results on exopher extrusion.

The Discussion is way too long and restates the same things at least 3 times in different words.

The following figures describe aggregates formation without having any connection to the topic/title of this paper and should be removed from this manuscript and put in another manuscript: Figure 3, Supplemental figure 1, 7, 8, 10, 11, 12, 15, 16, 20,

There is the potential for an interesting story to emerge from all these data, but it's not evident in the current version of the manuscript.

NCOMMS-23-01214-A

We thank Reviewers for their insightful comments and their appreciation of the work's significance. Reviewer input has helped us substantially improve the paper, both in substance and in message. We have taken all suggestions to heart and detail our response to review below. Reviewer comments are in blue, our responses are in black font.

REVIEWER COMMENTS

Reviewer #1 (Remarks to the Author):

The revisions are fine and address our main comments. The smaller points listed below could be considered by the authors for the final version.

--We thank Reviewer 1 for the insightful and useful comments and have tried to address all.

Some extra explanation:

*In the cell autonomous section of the paper (from line 242), it is shown that a single-copy GFP-tagged IFD-2 transgene, under its native promoter, can rescue the *ifd-2* ko phenotype on exopher formation. It is not clear why this was not also tested for *ifd-1*. The authors could comment on this in the manuscript.*

--We encountered some technical difficulties on the *ifd-1* deletion construction with natively expressed IFD-1; however we present rescue data on GFP::IFD-2 expressed natively in single copy (Figure 4D) as well as GFP::IFD-1 and mNG::IFD-2 expressed in single copy from touch neuron promoter (Figure 4-F) in their cognate deletion background. Additionally, we show that RFP::IFD-2 overexpressed from touch neuron promoter rescues the *ifd-1*(Δ) (Supplementary Figure 11A); so for historical reasons and in the context of 55+ data panels, we did not do native rescue for both *ifd-1* and *ifd-2* deletions; we point this out in the text: "We relied on single copy touch neuron-specific expression of GFP::*ifd-1* crossed to *ifd-1*(Δ) to confirm rescue of the exopher phenotype (Figure 4E).

*The intestinal expression of fluorescent tagged-IFD-2 shows a clear trend and an almost significant partial rescue in the exopher deficiency of the *ifd-2* ko worms (Supplementary figure 10). These results could very well be significant with increasing the n. Even when *ifd-2* is involved in exophogenesis acting from neurons, a small contribution from IFD-2 in exopher formation from the gut can't be rule with the current data. This should be briefly discussed in the text.*

--This is a good point on current Supplementary Figure 9 and we appreciate the suggestion to look more carefully at the intestinal rescue. The WT control vs the *ifd-2* deletion is P=0.0054; WT control vs. intestine rescue with mNeonGreen::IFD-2 is P=.07; the deletion vs the deletion+rescue is P=0.29; 5 repeated trials were performed, which is in line with the high n number we routinely measure throughout the paper. We agree on the trend for the comparison, and now state in the text:

"Comparing *ifd-2*(D) to *ifd-2*(D) P_{vha-6}mNeonGreen::IFD-2, we find that intestinal expression of mNeonGreen::IFD-2 does not rescue the neuronal exopher phenotype of *ifd-2*(Δ) (Supplementary Figure 9) consistent with a predominant role for *ifd-2* in exophogenesis in the neuron (Figure 4E-G) rather than in intestine, although a trend toward partial rescue leaves open the possibility of some intestinal contribution. Overall, touch neuron-specific rescue of the exopher production deficits in *ifd* null mutants support that IFD-1 and IFD-2 primarily act autonomously within the touch neuron to influence exopher production."

Small edits:

The x axis legends in Figure 1D and 1E for the rescue should denote "Pmec-7GFP::IFD-1" and "Pmec-mNeonGreen::IFD-2" to make more clear that these experiments were done with exclusive expression in touch neurons. Alternatively, this could be added in the figure's titles.

--We added the clarification that genetic rescue is a “single/high copy rescue from touch neuron promoter” with additional annotations added directly to the current main figure 4 and wrote out that the construct is “P_{mec-7}GFP::IF” in the figure legend title.

At the end of line 240, there was a statement in the previous version to acknowledge that other proteins than IFD-1 and 2 are likely involved in exopher production, giving that the double mutant is still able to produce them. It would be good to keep this statement: “Because some exopher production is evident when ifd-1 and ifd-2 are both absent, we infer that a redundant activity or a parallel pathway must also contribute to exopher formation.”

--We thank reviewer for this suggestion and added this double IF-mutant RNAi experiment (Supplementary Figure 8B) back in to the manuscript, along with a clarifying sentence added to the manuscript.

Main text: “Some exopher production remains evident when both *ifd-1* and *ifd-2* are absent, suggesting that additional IF proteins might influence exopher production. Indeed, when both IFDs are absent, RNAi knockdown of other *C. elegans* IF genes in a strain sensitized for touch neuron RNAi suggests that more IF proteins may modulate TN exophogenesis (Supplementary Figure 8B).”

Legend for Supplemental figure 8B includes the excerpted sentence:

“Because some exopher production is evident when *ifd-1* and *ifd-2* are both absent (Figure 4), we infer that a redundant activity or a parallel pathway must also contribute to exopher formation.”

In Supplementary Fig 8A, the legend says the merged comes from GFP, mCherry and DIC channels. The DIC channel is missing.

--We thank Reviewer 1 for this mention and have clarified and confirmed accuracy of the legends and figures. When we rearranged the manuscript and considered R2’s response on including excessive supplementary figures, we elected to take out previous Supplementary Figure 8 showing lack of visual expression of transgenic and endogenously expressed IFDs. We point out these negative data via mention in the text.

Also, giving the multiple single and multicopy controls throughout the text, which makes the story very difficult to follow in certain points, it would be good to add “OE” in the title of figures 8A and 8B.

--We added better indication of overexpression constructs used when necessary for exopher studies.

New figure labels to denote copy number for exopher studies:

Figure 4D-G, Figure 6E, Supplementary Figure 9, Supplementary Figure 10, Supplementary Figure 11, Supplementary Figure 13F, Supplementary Figure 15

Figure 8D shows endogenously tagged IFD-2 but the legends describes “ifd-1 gene ifd-1(bz477)”, please clarify this.

--We thank Reviewer 1 for catching this mistake. As noted above, we have removed previous Supplementary Figure 8 and only describe findings in the text.

The section starting from line 280, to test gene dosage effects does not correlate well with the subfigures shown in Supplementary Figure 6B.

--We significantly reduced the language describing the experimental interpretation in the main text and have verified correct mention of the figures in the manuscript.

The text describes first Sup. Fig 6B and the effect of ifd-2 OE into ifd-1 ko animals, next Sup. Fig 6C and finally Sup. Fig 6A. Please describe the data in a more logical order that correspond better with the figures or adapt the figure.

--We have gone through the text carefully to ensure that the figures are referred to in order.

Also, only the cross complementation of IFD-2 OE in the ifd-1 ko background is shown, but the vice versa mentioned in the text, which should be the IFD-1 OE in the ifd-2 ko background is missing. Is the ifd-1 ko; IFD-2 OE significantly different than the ifd-1 mutant in figure 6A?

--We edited the text to accurately and clearly reflect what we present in the figures. To clarify here, we report IFD-2 OE in the *ifd-1* deletion background in Supplementary Figure 11A. We do not present the reverse cross-complimentary experiment (with IFD-1 OE in *ifd-2*) but we do show that IFD-1 OE can rescue the double mutant, now presented in main Figure 4G.

The statement "Human neuronal intermediate filament protein hNFL can complement C. elegans ifd-2 deletion to promote exopher formation" should be rephrased as "can partially complement..."

--We revised the text to ensure that we highlight partial rescue.

Supplementary figure 22, although presented in the rebuttal letter to further support the neuronal effect of FTT-2, is not referred, or explained to in the text.

--Great to catch that these data did not stand out. We have verified that all figures are referred to, and explained, in the text.

Reviewer #2 (Remarks to the Author):

This manuscript is a mish-mash of experiments showing a multitude of factors regulating aggresome formation in neurons of *C. elegans*, but the manuscript is unfocused and instead of describing how IFs regulate exopher extrusion (the title of the paper) it meanders over various parameters of aggresome formation that while interesting should be removed from this paper and described in its own manuscript.

--The title: "*Intermediate Filaments Associate with Aggresome-like Structures in Proteostressed C. elegans Neurons and Influence Large Vesicle Extrusions as Exophers*" – documents two newly described functions of IFDs in proteostressed *C. elegans* touch neurons. We respectfully argue that both of these IFD functions are accurately portrayed in the title and documented in the paper. Implications of both facets of IFD involvement are novel and highly relevant to current mysteries in neurodegenerative disease. Since the linking of aggresomes and extrusion biology are featured in the title with emphasis on both facets in our revised manuscript, we did not significantly change the title.

The figures are similarly a mixture of information, often showing non-relevant things while omitting critical points. The approach seems to be to drown the reviewer in data, irrespective of whether they are or are not relevant to the story. Many critical figures have been relegated to supplemental, presumably to overcome the space limitations but, really 24 supplemental figures! Moreover, many of those figures are actually essential to the point of this manuscript while some of the figures or figure panels that are in the main manuscript don't add to the main thrust of the story.

--As noted above, we overhauled the data presentation from the previous submission, in part with attention to R2's concerns. In doing so, we carefully considered which experiments to include in the main text; and we have retained most of the supplemental data for interested readers. Space constraints as well as previous review comments have shaped contents (previous reviewer comments invited over 13 additional supplemental figures which includes 55+ panels added to the original submission). Our intention was not at all to drown reviewers in data, but rather to respond to reviewer concerns and document our findings most rigorously.

Currently there are 17 supplementary figures.

In its current form, the results and the writing are so confusing that in the end it's almost impossible to decipher what has been fully documented, i.e. which exopher cargoes have been completely analyzed to show how they

form exophers, whether they are surrounded or just adjacent to IFs, whether IFs and other regulators influence their expulsion from cells, and how the relative distribution of cargo and IFs within the cellular aggregates changes/doesn't change in the expelled exophers. Some of the data are probably dispersed in different figures and supplemental figures, but this reviewer had a hard time figuring out what exactly has been done and it is likely that readers would be similarly confused.

--Considering the reviewer and Editor's suggestions, we have changed the organization of the paper, separating data on aggresomes and following with data on exophers. We discuss polyQ and mCherry cargos separately, and in detail. We have worked hard on ensuring clarity in the revised presentation. We included a summary figure showing aggresome formation extrusion outcomes as Supplemental Figure 16. We thank reviewers and editors for the suggestion on the manuscript organization.

Additional issues:

1 Figure 1A – add what we are looking at - is that mCherry? And what is mCherry and why does it aggregate?

--We clarified the nature of the mCherry in the figure legends and we put an additional label directly in the figure, labeling "Pmec-4mCherry" for clarity, now Figure 4A.

We added the following to Figure 4A legends: "In general, several red fluorophores are known to aggregate in cells.⁹³ (PMID: 35846353). (We report here that the mCherry, which is highly expressed in the *bzIs166*[P_{mec}-4mCherry] strain, is useful as a touch neuron and exopher marker, but can concentrate in structures labelled with LMP-1::GFP and are thus presumptive lysosomes (See Supplementary Figure 5))."

We explain this biology and limit the reference to mCherry as aggregated in the revised manuscript.

1B) should be removed and replaced with a series of IFs/EMs/movies showing the formation of an exopher; should show IFs with mCherry – do the IFs surround the aggregate or are interspersed with the cargo or something else? – this should be shown for all exopher cargo described in this manuscript

--We use the cartoon in previous Figure 1B of an exopher budding (now Figure 4B) to introduce the exophogenesis process to the generalized audience, depicting clearly defined subcellular entities and summarizing "stages" identifiable by light microscopy. Adding information to this cartoon regarding IF localization in the process of exopher-genesis at the introduction point would not make sense, as those are data we document in subsequent figures.

-Expansion of electron microscopy (EM) data is beyond the scope of the paper and the finances supporting our work.

-Cargo notes. In Supplementary Figure 5 and in a separate text section we more clearly make the point that although the mCherry is clearly an exopher cargo, the concentrated mCherry is 'stored' in a lysosome-like compartment and not at the aggresome. This is in contrast HttPolyQ aggregates, which are concentrated at the aggresome compartment (Figure 3D-F, Supplementary Figure 4). In sum, different storage for two definitive exopher cargos; a bit complicated, but we document with rigor that is the operative biology.

We do include and discuss the polyQ and IFD relationship over time (Supplementary Figure 4). This is novel *in vivo* data in the field and likely important information on formulating working models of aggregate biology in high proteo-stress cells.

We also show that IFD punctae do get out in exophers ~15-50% of exopher events (Previous Supplementary Figure 18 and 19A-B, now main Figure 5), implicating aggresome elimination as a mechanism for clearance of these structures.

We agree that cargo dynamics and interaction with IFDs during exophogenesis is certainly interesting but currently those extensive studies on low frequency events remain beyond the scope of this paper.

2. Figure 2 – A-C and E-K are irrelevant and should be in another paper

--We respectfully argue that previous Figure 2A-C (Current Figure 1A-C), which characterize the main fluorescent reporters used in our study, report on biology relevant to this manuscript. We hope the reorganization of data makes this point more clear.

Previous Figure 2E-K (Current Figure 1E-K) rule out that the IFD proteins localize to known organelles and support their distinctive subcellular localization, important data relative to aggresome analysis *in vivo* in a genetic model of proteostress.

3. Figure 3 – completely irrelevant to this paper and should be moved into another paper on aggresome formation

--Previous Figure 3 reported on conditions and genetic requirements for IFD concentration into aggresome-like structures. Involvement of IFs in any aggresome-decoration biology had not been previously described in *C. elegans* neurons, and the characterization is important to anchor studies of aggresome biology in an *in vivo* context moving forward. In our revised manuscript, previous Figure 3 (on MTs, dynein requirement on IFD collection and FTT-2/HSP-1 colocalization to the IFD site) is in main Figure 2. Again, the importance of these data to the story are now better emphasized with the changed order of data presentation.

4. Figure 4 – A and B should be in Supplemental; must show exophers with this cargo in a figure; should also show the Htt cargo in this figure and document and quantitate exophers (with or without IFs) containing Htt in this figure.

--To summarize,

Previous Figure 4A-B shows IF colocalization with UBQ-2 ubiquitin. (Now Figure 3A-B).

Previous Figure 4C-E shows IF colocalization with HttPolyQ. (Now Figure 3C-F).

We show IFD+Q74 colocalized in an exopher in previous Supplementary Figure 18 (Now Figure 5D) (and quantitate these features, Figure 5E-F).

Regarding the request that previous Figure 4A-B, on UBQ-2 colocalization with IFDs, should be in the supplemental, we point out that this figure, as is, serves to document IF interaction with potential aggregate components (PolyQ aggregates themselves or ubiquitin, a critical feature of aggresome-like organelles), and we keep the valuable study as a main figure panel in current Figure 3.

Reviewer 2 also asks for a new figure showing and quantitating UBQ and Htt cargo in exophers. We had addressed IFs and the percentage of IFs containing specific Htt cargo that end up in exophers (previous Supplementary Figure 18, now a main figure 5). Our assessment is that additional exopherogenesis counts +/- IFD, +/- specific Htt or UBQ cargo would require a significant amount of time for strain construction and evaluation without contributing a critical extension to the story, so we have not included these studies in the revised manuscript.

5. Figure 5 – B is irrelevant and fits into another paper on aggresome formation.

--Previous Figure 5B shows human intermediate filament (hNFL) colocalization with UBQ-2 (requested in previous review; now Figure 6), mirroring *C. elegans* intermediate filament colocalization with ubiquitin (Previous Figure 4A-B, now in Figure 3).

The data independently support that human NFL acts similarly to *C. elegans* IFDs in touch neurons, and localize to a UBQ-concentrated organelles. This is important information on conserved aggresome biology and thus we left NFL colocalization with UBQ in main Figure 6.

Figure 5 D – label what is red and what is green; why are these IFs not surrounding the cargo in the exopher—when did the remodeling occur since they were together in the cell in Figure 5C?

--Previous Figure 5D (Now Figure 6D) shows a picture of mNG::NFL puncta in the soma and in the exopher in a P_{mec-4} mCherry background. We added a genotype label to Figure 6D, which should clear up potential confusion.

We do not refer to remodeling in this figure—and now make clear the images are static and not from a time course study. Panels A, B, C, and D, are different (static) images from different strains, clearly noted in legends and strain table 1. That the image in previous Figure 5D (Now Figure 6D) is not related to the cells shown in the other panels, is now stated clearly in the legend.

Include in the paper in the appropriate place the data from supplemental figures 5, 14, 17, 18, 19, 21, and 22 – those are actually the interesting and informative results on exopher extrusion.

--The data we present in main vs. supplemental text is somewhat dictated by the revised order that separates out aggresome and exophers. We have shuffled some as requested, but since the paper has a different slant, we have settled on what we feel is a distribution that does the job of telling the two stories. We are comforted by the fact that supplement data are fully accessible to readers who want to consider data details.

The Discussion is way too long and restates the same things at least 3 times in different words.

--We have made significant revisions and cuts to focus the discussion.

The following figures describe aggresome formation without having any connection to the topic/title of this paper and should be removed from this manuscript and put in another manuscript: Figure 3, Supplemental figure 1, 7, 8, 10, 11, 12, 15, 16, 20.

--We argue that this paper reports on both roles of IFs in aggresome-like biology and in exophogenesis (in separate substories), and now emphasize this more clearly for the included data.

There is the potential for an interesting story to emerge from all these data, but it's not evident in the current version of the manuscript.

We thank R2 for comments that have helped us to revise and clarify data presentation. The paper is now much stronger for that.